# High-throughput triazole-based combinatorial click chemistry for the synthesis and identification of functional metal complexes

David R. Husbands [1], Çağrı Özsan[1,2], Athi Welsh [1,2], Richard J. Gammons [1] & Angelo Frei [1,2] ✉

Transition metal complexes have found many applications in the modern world ranging from catalysts, luminescent materials to bioactive compounds and more. However, methods for the systematic synthesis and evaluation of large numbers of metal compounds at a time are still limited. Here, a high-throughput combinatorial approach using copper(I)-catalyzed alkyne–azide cycloaddition chemistry is implemented to make 192 bidentate pyridyl-1,2,3-triazole ligands. This ligand library is coordinated to five metal scaffolds under mild conditions to yield 672 metal compounds, allowing for an accelerated exploration of the transition metal complex chemical space. The prepared libraries are showcased for compound discovery in antimicrobial applications through a "Direct-to-Biology" approach, with selected libraries further tested for catalytic and photophysical properties. Six promising metalloantibiotics are identified and re-synthesized, exhibiting activity against Gram-positive bacteria in the nanomolar range with favorable therapeutic windows. Two iridium complexes are selected from a transfer hydrogenation screening assay and re-synthesized, with one showing high catalytic activity. This methodology represents a significant step forward in the field of metal-based combinatorial chemistry and its application towards the systematic coverage of vast chemical spaces in the search for molecules with optimized properties.

The Copper(I)-catalyzed Alkyne–Azide cycloaddition (CuAAc Click) reaction to form substituted 1,2,3-triazoles (hereafter referred to as "triazoles") is one of the cornerstones of modern chemistry[1–4]. Functional group tolerance, bio-compatible coupling conditions and high yields make it a powerful transformation. Although mostly used as a convenient way of linking molecular fragments together, it is possible for the resulting triazole unit to bind to metal centers as a ligand. This coordination opens up avenues for catalysis and complexes with highly tunable photophysical properties[5,6]. Incorporation of a pyridine or other lone pair donating group to the triazole compound enables formation of bidentate ligands, which are able to chelate metal centers effectively via 5- and 6-membered metallocycles (Fig. 1)[7]. The binding modes of these ligands can be varied by switching the substitution site on the triazole ring (through different pyridine/azide structures), giving ligands with substantially different chemistries[8]. Coordination tends to occur through the more electron rich N3, giving more stable complexes, while coordination through the more electron deficient N2 is much rarer and can cause instability in complexes (Fig. 1)[9]. Triazoles

[1]Department of Chemistry, University of York, York, UK. [2]Department of Chemistry, Biochemistry & Pharmaceutical Sciences, University of Bern, Bern, Switzerland. ✉e-mail: angelo.frei@york.ac.uk

**Ligand design**

**Fig. 1 | Pyridine-triazole ligand design and metal coordination, and previously studied complexes featuring these ligands.** (Top) pyridyl-triazole ligand potential building blocks, numbering convention used in this paper and coordination modes to metal centres[8]. 1-R-1,2,3-triazole-4-(2-pyridine) = Tz-4-P, 4-R-1,2,3-triazole-1-(2-pyridine) = Tz-1-P, 4-R-1,2,3-triazole-1-(2-methylpyridine) = Tz-1-MP. (Bottom) Previous examples of pyridyl-triazole metal complexes for a range of applications; **A** Re-based sulfonamide anchored complex[15]; **B** tris(homoleptic) Ru complexes[16]; **C** tis(homoleptic) mer-Os complex[17]; **D** Mn(CO)₃-Clotrimazole complex[19]; **E** RuCy complex with a coordinated sugar derivative[20]; **F** cyclometallated IrCN complex[21].

can also coordinate to metals as *N*-heterocyclic carbenes rather than through the N donors[10].

From our perspective, the coordination chemistry of these triazole ligands holds great promise for the development of metal-based antibiotics[11]. This area is becoming increasingly relevant due to the rise of antimicrobial resistance (AMR) to the current arsenal of antibiotics, which has been designated by the WHO as one of the top global health threats[12,13]. Concurrently, there are very few antibiotics being developed, with only 19 antibacterial drugs being approved for use between 2013 and 2023, and of these only 2 represent a new chemical class[14].

There have been several examples of bidentate pyridyl-triazole ligands being used for therapeutic applications, from using CuAAc-formed triazoles to anchor sulfonamide drugs to metal centers (increasing activity against methicillin-resistant *Staphylococcus aureus* (MRSA), Fig. 1A)[15], to using bidentate triazole ligands directly in forming tris-homoleptic ruthenium [Ru(L)₃]²⁺ complexes (showing broad spectrum antibiotic properties, Fig. 1B)[16], tris-homoleptic osmium *mer*-[Os(L)₃]²⁺ complexes (photoactive against MRSA, Fig. 1C)[17] and rhenium(I) carbonyl complexes (displaying activity *vs.* both Gram-positive and Gram-negative bacteria)[18]. Additionally, similar bidentate pyridyl-triazoles have been coordinated to a manganese(I) tricarbonyl scaffold, utilizing Clotrimazole as an axial ligand, and were demonstrated to be active against Gram-negative bacteria (Fig. 1D)[19]. In terms of other therapeutical applications, a ruthenium(II) arene (Fig. 1E) scaffold was coordinated to derivatized sugar pyridyl-triazole ligands (selective activity against ovarian cancer cells)[20], and pyridyl-triazole ligands were used to coordinate cyclometallated iridium(III) scaffolds, generating several compounds with photoluminescence properties (Fig. 1F)[21]. These could be relevant for photodynamic therapy applications against either bacteria or cancer cells, similar to recent work on cyclometallated complexes by Kench et al.[22,23]. Platinum compounds containing triazole ligands or linkers have also been investigated for anticancer therapy[24]. Although an impressive array of compounds has been made utilizing triazole ligands[25], it should be noted is that in almost all cases, only a handful of individual complexes have been synthesized and tested for their intended applications. One of the reasons for this is the synthesis of the triazole ligand, which requires an azide to be generated as a precursor. Azide formation typically occurs in batch via a reaction between sodium azide and an aryl/alkyl bromide, requiring forcing conditions and undesirable solvents, and can generate significant impurities[26,27].

Recent developments by Meng et al.[28] have opened the door to high-throughput synthesis of this class of ligands through sulfur(VI) fluoride exchange (SuFEx) chemistry under ambient conditions, efficiently generating azides in situ from virtually any primary amine (Fig. 2). We envisaged that this methodology could be used to generate substantial and varied libraries of pyridyl-triazole based ligands. Notably, these ligands offer stability benefits over Schiff-base type counterparts, which are prone to hydrolysis and degradation in the presence of acid or base. Coordination of this library to various metal scaffolds in a similar vein to previous work by our group and others using combinatorial chemistry approaches with Schiff-base ligands[29–31] would allow us to create libraries of complexes. These could in turn be screened for antibacterial hits, toxicity towards mammalian cells and other potential areas of discovery such as catalysis and photophysical applications.

To rapidly cover and focus in on the vast chemical space represented by this chemistry, we have turned to a "Direct-to-Biology" screening approach, which has been successfully used across academia and industry[22,32–41]. This approach involves the high-throughput synthesis and characterization of compound libraries, followed by biological testing of reaction crudes without purification. Concurrently, the reaction components (in our case metal scaffolds and ligands) are tested to support that any observed biological activity is likely from the putative metal complex. There are potential issues with

impurities having an outsized effect in these assays, so high yielding reactions, such as those typically used in combinatorial chemistry are favored[29–31]. As a result, while it is possible to get false positives with a Direct-to-Biology approach, false negatives are rare. The final step is hit validation, where target compounds which showed promising properties in the crude-screening are re-synthesized and purified, and finally re-tested to confirm their biological activity. Overall, Direct-to-Biology represents a powerful screening technique that allows for coverage of a large amount of chemical space quickly, reduces time wasted in making inactive compounds, and funnels down to a small number of compounds with desirable properties. A similar approach can also be taken with photophysical and catalytic testing[42,43].

## Results and discussion
### Preparation of pyridine-triazole ligand libraries

We began by optimizing the conditions described by Meng et al.[28] for the in situ generation of azides from primary amines, followed by CuAAc Click reactions with 2-alkynepyridines. Initially, significant alcohol impurities were observed, but this was remedied by changing the copper ligand from tris-hydroxypropyltriazolylmethylamine (THPTA) to tris(2-benzimidazolylmethyl)amine (BimH)₃. Additionally, we found that the reported phosphate buffer was not required in our system, and low catalyst/ligand loadings of 2.5 mol% were sufficient. With the optimized conditions in hand, we used an Opentrons liquid handling robot to generate a library of 96 triazole-4-pyridine (Tz-4-P) compounds (24x amines, 4×2-alkynepyridines, Fig. 3A). The success of these couplings was verified by Liquid Chromatography coupled Mass Spectrometry (LC-MS, Fig. 4). A further library of 96 triazole-1-methylpyridine (Tz-1-MP) compounds (12x alkynes, 8×2-picolylamines, Fig. 3B) was prepared using the same method. Under the same conditions, we also attempted to couple 2-aminopyridine with alkynes to generate a Tz-1-P library, but this was unsuccessful.

The average conversion efficiency (as determined by automated LC-MS chromatogram integration at 254 nm)[44] was 75 ± 16%, showing that despite the structural diversity of the amines a generally high conversion efficiency was observed (Fig. 4). In terms of trends across the formation of this library, the electron withdrawing ester group on M11 tended to lead to lower conversions (both for the initial click reaction, and for subsequent coordination), although other electron deficient amines (M13, M17) seem to form ligands in high yielding reactions comparable to electron rich systems (M12, M19). As such, it does not appear that the electronics of the amine building blocks have a significant effect on the outcome of the initial CuAAc reaction. The LC-MS traces of ligands containing 7-aminocephalosporanic acid and ampicillin (M15, M16, respectively) were not as clean as the others, and it was found that these ligands and their metal compounds slowly degrade over time in solution (See Supporting Information (SI) Figs. S24, 25). However, it was decided to include them in further testing as they represented more complex structures with β-lactam motifs, and ideally the coordination of these derivatives to metal centers would enhance their biological effects[45]. On a practical level, it was discovered that several of the target ligands were poorly soluble in DMSO and tended to precipitate or crystallize out of solution. This was especially prevalent among Y4 ligands, where the pyridyl bromine group seems to have significantly affected solubility. This was remedied by pulverizing the affected samples through sonication and aspirating the mixture to homogenize it before dispensing the ligands for further reactions.

Similarly, the Tz-1-MP library was synthesized with a high average conversion efficiency (77 ± 12 %), with no noticeable degradation of the products (Fig. 5). Initial tests included two aliphatic alkynes (cyclohexylacetylene and cyclopropylacetylene), but the products of these reactions were not observable by LC-MS. As these alkynes also exhibited problematic volatility, the building blocks were restricted to aromatic alkynes only. These were in turn limited to relatively simple

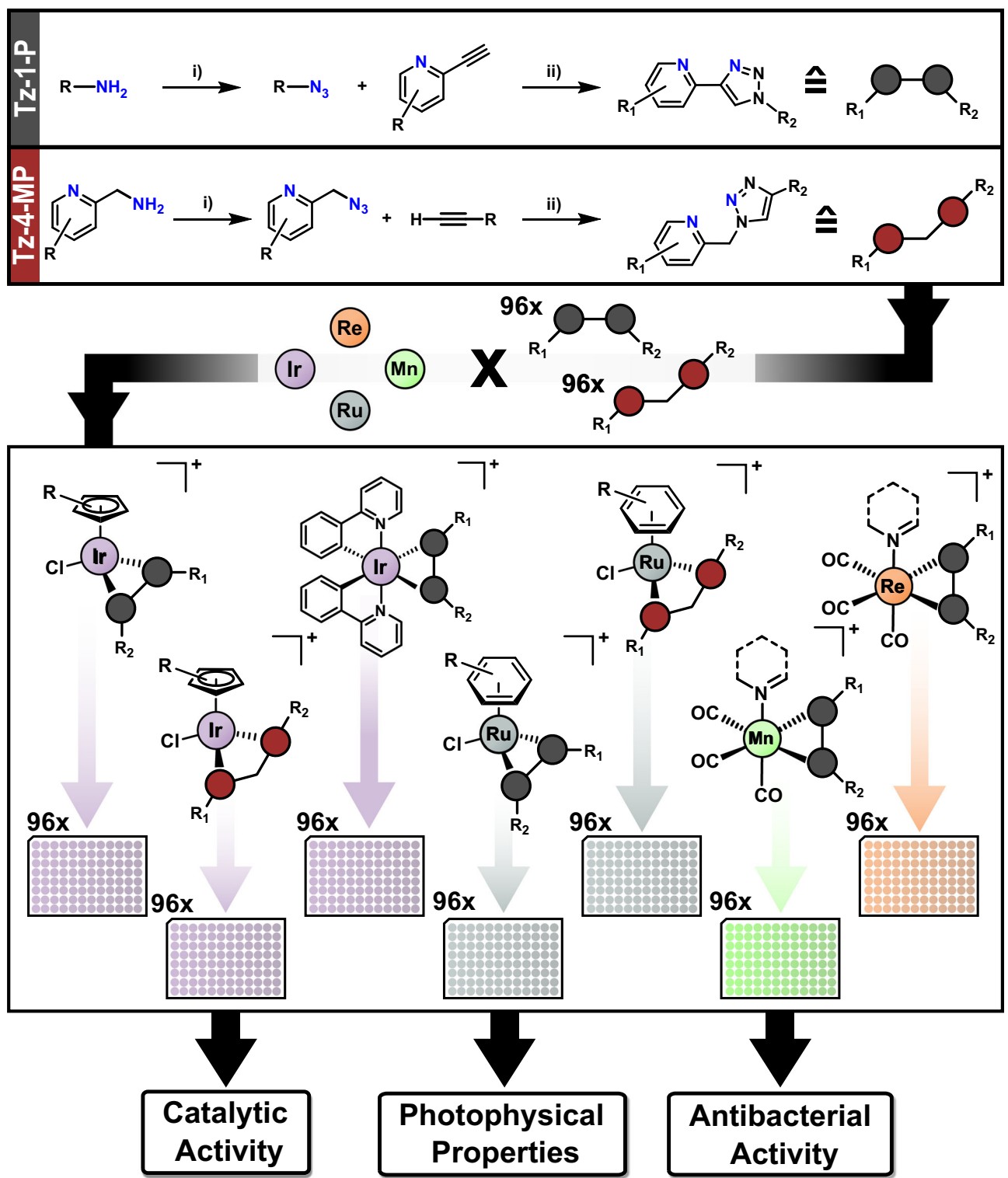

**Fig. 2 | Overview of combinatorial methodology for the generation and testing of metal complex libraries.** Conditions for i) Amine (80 mM) FSO$_2$N$_3$ (1.05 equiv.), KHCO$_3$ (4 equiv.), MTBE/DMSO/H$_2$O (11:84:5), r.t., 18 h; conditions for ii) Alkyne (80 mM, 1 equiv.), sodium ascorbate (0.5 equiv), CuSO$_4$ (2.5 mol%), (BimH)$_3$ (2.5 mol%), DMSO/H$_2$O (3:1), r.t., 18 h; to give the triazole library at 20 mM concentration (for more details, see Supporting Information Section 2.1). Details of the coordination of the triazole libraries with metal can be found in Fig. 3.

derivatives, which demonstrates a limitation of using aromatic alkynes (Tz-1-MP) compared to primary amines (Tz-4-P) for triazole formation. Conversely, the use of picoline-derived amines allowed for a substantially larger variety in the pyridine fragment of the ligand compared to the Tz-4-P library, with several halogenated derivatives included. Although the structural variety of the Tz-1-MP library is not as broad as Tz-4-P, the overall conversion is comparable, with only A3 (highly electron-deficient *p*-nitro group) and P4 showing consistently lower conversions. No solubility issues were observed with the Tz-1-MP library.

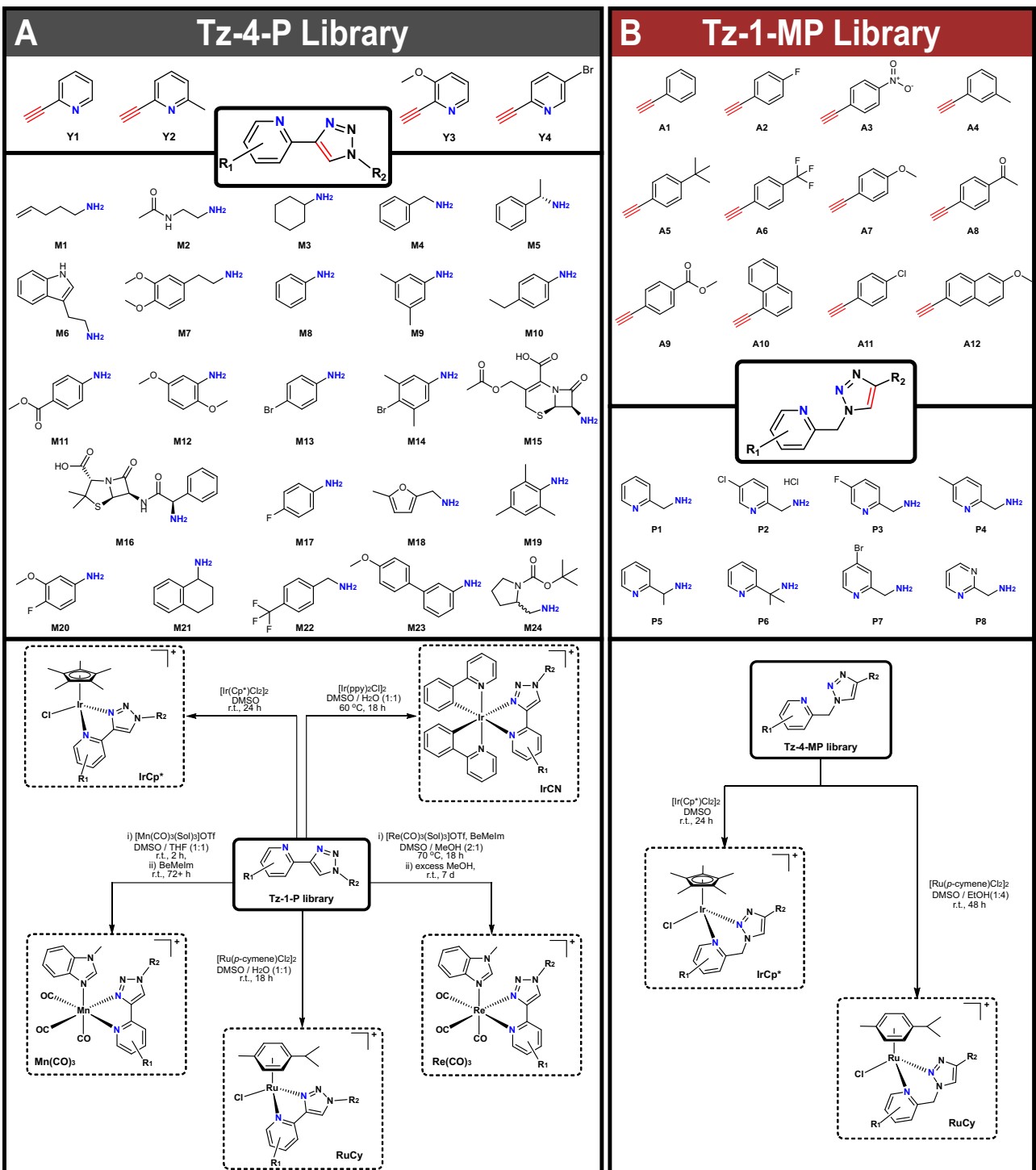

**Fig. 3 | Amine and alkyne building blocks forming the ligand library, metal coordination conditions.** Overview of amine and alkyne building blocks utilized for the Tz-4-P (**A**) and Tz-1-MP (**B**) libraries and synthetic schemes and conditions for the coordination of Tz-4-P and Tz-1-MP libraries to metal scaffolds to form the metal complex libraries. Sol = solvent, Cp* = pentamethylcyclopentadienyl, BeMeIm = 1-methylbenzimidazole, ppy = 2-phenylpyridine.

## Preparation of metal complex libraries

After test coordination reactions using M4Y1 to optimize conditions, the entire Tz-4-P library was coordinated to the following metal scaffolds (Fig. 3A): [Ir(ppy)₂Cl]₂ (ppy = 2-phenylpyridine, IrCN), [IrCp*Cl₂]₂ (IrCp*), [Ru(p-cymene)Cl₂]₂ (RuCy), Re(CO)₅Cl (Re(CO)₃) and Mn(CO)₅Br (Mn(CO)₃). These scaffolds were chosen due to their previously demonstrated biological activities and their amenability to combinatorial chemistry[19,23,30,31]. Both manganese and rhenium scaffolds were pre-activated with silver triflate, and solvent libraries

without 1-benzyl-2-methylimidazole (BeMeIm) as the axial ligand were also generated as controls. The reaction mixtures for the five different metal complex libraries were set up simultaneously in a semi-automated high-throughput manner using the Opentrons OT2 liquid handling robot.

Coordination of the Tz-4-P library to the IrCN scaffold was achieved under conditions far less forcing than current literature procedures, which require toxic, high boiling point solvents, temperatures >150 °C, or microwave conditions[22,42]. In our procedure, high

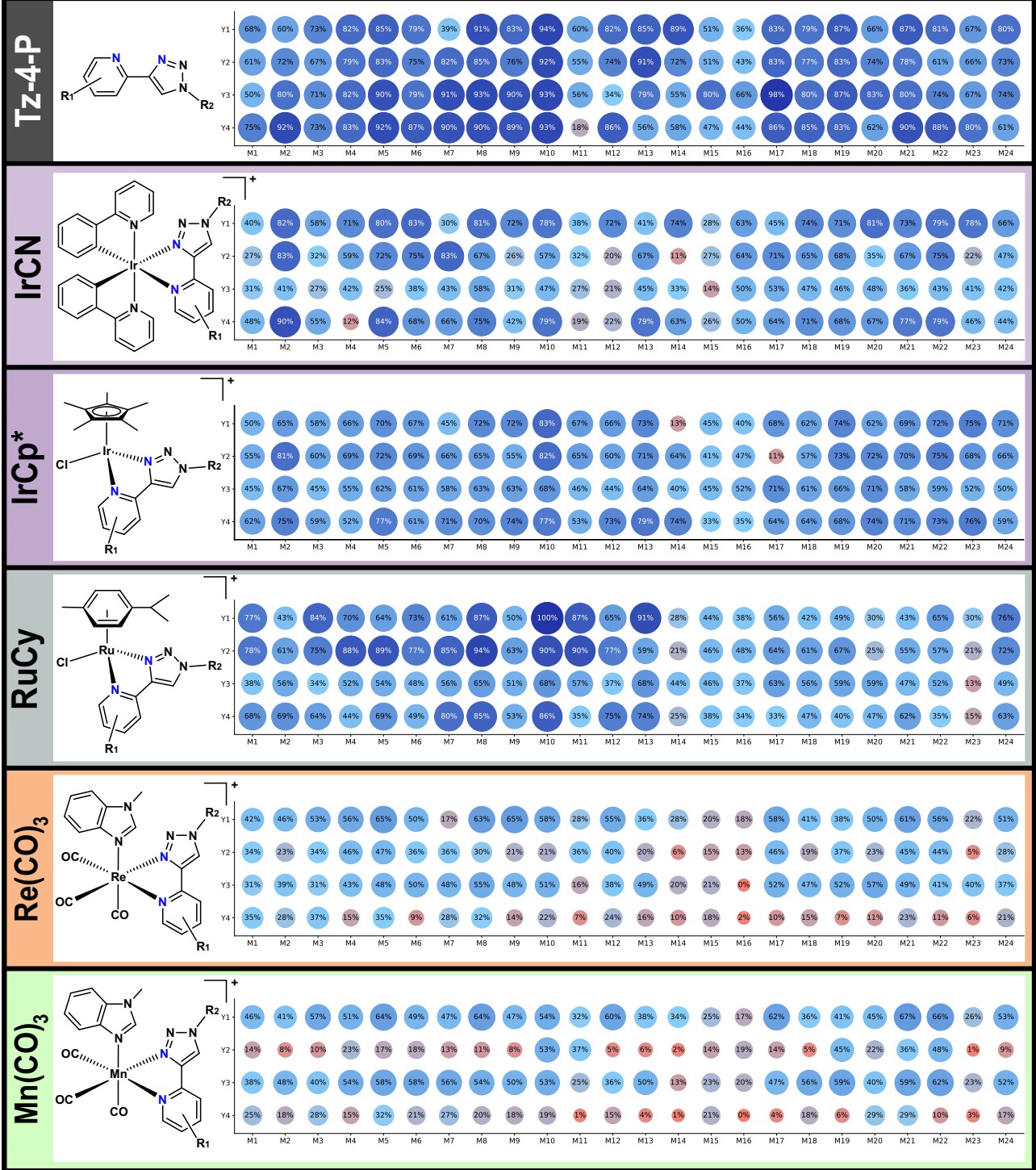

**Fig. 4 | Conversion to product of the Tz-4-P ligand and metal complex libraries.** Purity as ascertained by peak area% of the total from LC-MS data (254 nm). Compound peaks were identified automatically using the mass of the target ion and the corresponding UV peak was automatically integrated by custom Python scripts (Supporting Information Section 2.9). RuCy library conversions were normalized to 100% due to low UV response at 254 nm, and the conversions of IrCp* libraries is an aggregate of the parent ion (M+) and the MeCN adduct (M2+) peaks.

conversion was achieved by heating the ligands and [Ir(ppy)$_2$Cl]$_2$ dimer in a 1:1 mixture of DMSO:H$_2$O at 60 °C for 18 h. IrCp* and RuCy libraries were generated in good purity under ambient conditions. The IrCp* library is being explored in an ongoing project for biological activity, so will not be discussed further here.

For the Re(CO)$_3$ library, poor conversion to the target complexes was initially obtained (1:2 MeOH/DMSO, 70 °C, 18 h). After addition of an excess of MeOH, the reaction crudes were left standing for one week at room temperature to allow for the solvent to evaporate, giving significantly improved product formation. This is likely due to DMSO solvent competitively binding with the rhenium scaffold instead of BeMeIm. A similar effect was observed for the Mn(CO)$_3$ library, as when conversion is low, the byproduct is often the solvent adduct (e.g. Mn(CO)$_3$M2Y3, see Supporting Information Figs. S26, 27). Previously,

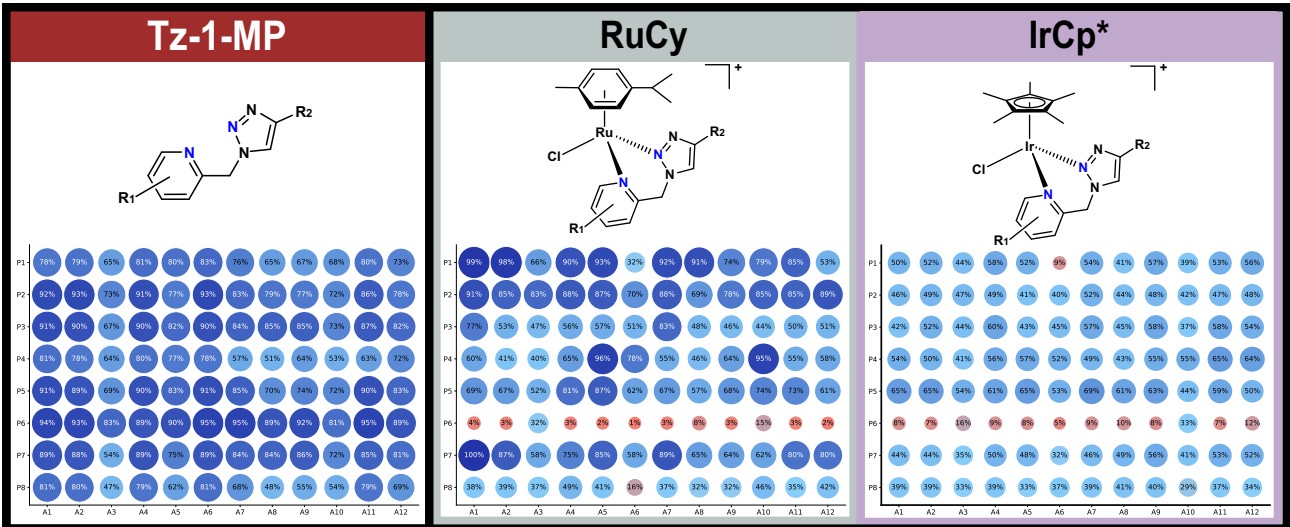

**Fig. 5 | Conversion to product of the Tz-1-MP ligand and metal complex libraries.** Purity as ascertained by peak area% of the total from LC-MS data (254 nm). Compound peaks were identified automatically using the mass of the target ion and the corresponding UV peak was automatically integrated by custom Python scripts (Supporting Information Section 2.9). RuCy library conversions were normalized to 100% due to low UV response at 254 nm, and the conversions of IrCp* library is an aggregate of the parent ion (M$^+$) and the MeCN adduct (M$^{2+}$) peaks.

our group has shown that manganese(I) carbonyl complexes with solvent as the axial ligand tend to have low antibacterial activity[31]. Manganese(I) tricarbonyl complexes are often light-sensitive, so this library was prepared, handled, tested and stored in the dark.

Test coordination to the same scaffolds using Tz-1-MP (P1A1) implied that DMSO bound competitively with all metal centers, preventing coordination in many situations, even at elevated temperatures. The exception to this was the [IrCp*Cl$_2$]$_2$ scaffold, which yielded product and led to the creation of an IrCp* library (Fig. 3B). In addition, excess EtOH with [Ru(p-cymene)Cl$_2$]$_2$ gave the desired complexation with P1A1 in high yields, so the corresponding RuCy library was prepared with the Tz-1-MP ligands (Fig. 3B).

The poor complexation of the Tz-1-MP library could be due to the greater flexibility imparted by the methyl linker, forming 6-membered metallocycles rather than 5-membered metallocycles formed by the more rigid Tz-4-P library. Generally, rigid 5-membered rings are significantly more stable than 6-membered rings due to fewer degrees of freedom and a more optimum "bite angle" (the angle between the donor groups and the metal center)[46]. More specific examples involving Tz-1-MP scaffolds from Urankar et al. show that under certain coordination conditions, these ligands can bind in a monodentate fashion through the triazole ring (to palladium) rather than a chelating effect[47]. Maisonial et al. similarly demonstrated that for triazoles with a pendant primary amine, 5-membered chelates were significantly more stable when coordinated to platinum than 6-membered metallocycles[48].

Typically for metal-based combinatorial synthesis, libraries are only partially characterized (either random representative samples or predetermined subsets, e.g. using $^{19}$F NMR for fluorine containing compounds) to reduce the time-consuming characterization step to a more practical level[29,42,43]. In our case, the charged nature of the metal complexes makes them amenable to characterization by LC-MS, a technique that itself can be highly automated both in collection and assessment of the data (see Supporting Information Section 2.9). Therefore, the purity and identity of all compounds were assessed and the processed LC-MS spectra, target complex percentage conversion and retention times are supplied as Supplementary Data 1 with this article.

For Mn(CO)$_3$, and to a lesser extent the Re(CO)$_3$ library, Y2 containing ligands were consistently poor at coordination. This could be

due to steric clashing of the pyridine methyl group with the axial ligands during coordination. Similarly, Y4 containing ligands performed poorly, possibly due to the lower solubility of the ligands, or electron withdrawing effect of the pyridyl *para*-bromine. Generally, ligands formed from simple aryl amines such as M8, M10 and M17 tended to coordinate most efficiently. Our data also indicates that M group sterics do not affect the yields, likely due to these groups being spatially removed from the metal center. Electronic effects seem to be more consequential, with strong electron donating units such as methoxy- and methyl-groups favored. Aliphatic groups from building blocks such as M1, M2 and M3 also coordinated well, again implying the prevalence of electronic effects on coordination over steric influences.

For the Tz-1-MP library, coordination of the P6 variants was consistently poor, likely due to the *gem*-dimethyl groups on the methyl linker. As well as being a useful functionality in medicinal chemistry[49], it was envisaged that the Thorpe-Ingold effect would be beneficial to coordination, both increasing rotational barriers and decreasing the bite angle of the ligand[50]. Instead, it appears that the steric bulk of the *gem*-dimethyl groups effectively inhibits chelation in P6 containing ligands. In most other cases, coordination of this library to the metal centers was reasonably successful, and the general high yielding formation of this ligand library by CuAAc reaction is a testament to the robustness of the optimized Click conditions.

## Photophysical evaluation and catalytic testing

Taking inspiration from the work by Kench et al., we investigated the photophysical properties of the crude IrCN library with the aim of exploring the possibility of photodynamic therapy applications[22]. A range of fluorescence intensities was observed, but the maximum emission wavelength ($\lambda_{max}$) was evidently dependent on the Y alkyne substituent (see Supporting Information, Fig. S18). This indicates that chemical modifications on the pyridine (Y) are generally more electronically impactful than the amine modifications (M, affecting the triazole). Information about the excited state identity and lifetime can be inferred from the shape of the emission spectra[42]. The typical excitation and decay cycle of cyclometallated iridium(III) complexes involves excitation from the ground state to a singlet metal/ligand-ligand charge transfer excited state (MLLCT, d$\pi \to \pi^*$ transition), followed by intersystem crossing to a triplet $^3$MLLCT excited state. The

decay of this pure excited state is often rapid and is a single peak with a roughly Gaussian distribution, such as observed in Y2 and Y4 complexes. Mixing of the $^3$MLLCT state with a ligand-centered triplet state ($^3$LC) can also occur, giving a longer-lived excited state. This is characterized by a peak with a significant shoulder (Y1 and Y3). The longevity of the excited state can have significant impacts on the photophysical properties of molecules, including on the generation of reactive oxygen species (ROS).

To ascertain whether ROS was generated by the IrCN library, irradiation at 405 nm with ROS scavengers (ABDA (anthracenediyl-bis(methylene) dimalonic acid) and RNO/histidine (p-nitrosodimethylaniline)) was undertaken. ABDA was observed to degrade in the DMSO:PBS media, a previously reported effect upon irradiation below 450 nm[51]. Significantly less ROS was detected for all compounds compared to the Ru(bpy)$_3$$^{2+}$ (bpy = 2,2'-bipyridyl) control using the RNO/histidine assay[52]. This suggests that ROS generation for these IrCN compounds is occurring at a lower rate. Altering the cyclometallated ligand may result in complexes with higher ROS-production rates.

Following combinatorial work on ruthenium arene Schiff-base complexes by Weng et al., we tested the crude RuCy libraries for transfer hydrogenation (TH) catalysis using the Coumarin-N$_3$ assay developed by the same group[43]. As Coumarin-N$_3$ is hydrogenated to the amine, it becomes fluorescent, so the formation of the product can be monitored over time (Fig. 6A). The rationale behind this reaction is to take advantage of high formate concentrations in E. coli and S. aureus to hydrogenate azide pro-drugs in bacteria[53]. As can be seen in Fig. 6B, C, which shows the change in fluorescence ($\Delta_{fl}$) intensity from TH catalyzed by the RuCy libraries, RuCy(Tz-4-P) type compounds (Fig. 6C) are generally more active than the RuCy(Tz-1-MP) library (Fig. 6B), with 3 compounds (M11Y2, M13Y2, M24Y2) all having a $\Delta_{fl} > 15000$ arb. units, comparable with active compounds reported by the Ang group. Interestingly, the most active compounds all contain the Y2 motif with an electron donating methyl group in the 2-position, which seems crucial for activity.

We additionally tested the crude IrCp* libraries for transfer hydrogenation in an assay adapted from Miller et al. for evaluating the activity of iridium-based artificial metalloenzymes using harmaline (Fig. 6D–F)[54]. Iridium-based TH systems tend to be the most active compared to other transition metal catalysts[55], and for the harmaline assay this was demonstrated to be the case. Preliminary testing with the RuCy(Tz-1-MP) library showed no activity, while several of the IrCp*(Tz-4-P) (Fig. 6F) complexes showed high conversions of the harmaline starting material to product (measured by a change in absorbance ($\Delta_{abs}$) at 384 nm corresponding to harmaline consumption, confirmed by LC-MS, Figs. S10, 11). In the cases where good conversions were observed, the electron donating methyl and methoxy groups (Y2, M14, M23) as well as bromine groups were present (Y4). This is in line with previous work in the field where small changes in ligands can have dramatic effects on the activity of the catalyst[56]. This can be ascribed to a complicated interplay of steps in TH, which first require abstraction of the coordinating chloride, formation of a hydride species (favored by electron rich ligands)[57], substrate coordination (aided by labile ligands, often electron poor)[58] and concurrent transfer of hydride (enhanced by electron rich ligands). Upon testing the IrCp*(Tz-1-MP) library (Fig. 6E), it became apparent that these catalysts were significantly more active than their Tz-4-P counterparts. As 6-membered chelates, they are more labile while remaining electron donating, which could explain the higher catalytic activity[47,48]. Additionally, the IrCp*(Tz-1-MP) library tended to have a greater proportion of MeCN adduct (M$^{2+}$) peaks in the LC-MS traces than the IrCp*(Tz-4-P) library, indicating that halide abstraction is easier.

As such, we decided to resynthesize and re-test two of the most active complexes, IrCp*(P1A7) and IrCp*(P4A11). The lability of 6-membered chelates meant that for both compounds, some free ligand was observed in solution by NMR spectroscopy even after

purification. In the harmaline assay, only IrCp*(P4A11) demonstrated activity, on par with its crude. Upon examination of the crude LCMS of IrCp*(P1A7), there were a couple of significant impurities including free ligand and an unknown iridium(III) dimer, suggesting that the parent complex was not the catalytically relevant species. In contrast, the crude LCMS of IrCp*(P4A11) only showed the parent complex and the [solvato-iridium]$^{2+}$ species (See Supporting Information Section 2.6 for LC-MS traces and kinetic graphs). Interestingly, both purified catalysts were active in the coumarin-N$_3$ assay, with IrCp*(P4A11) displaying superior activity. Both catalysts surpassed the activity of the best crude RuCy catalysts (Table 1, $\Delta_{fl}$ = 46056, 31295, 16505 arb. units for IrCp*(P4A11), IrCp*(P1A7) and RuCy(M13Y2), respectively)

We further investigated the effect of exogenous chloride by testing the two pure catalysts in PBS 1X / H$_2$O co-solvents with $^t$BuOH, with and without halide extraction by AgOTf (Fig. 6G–I). As can be seen in Table 1, removing excess chloride accelerates the reaction in all cases, likely due to biasing the equilibrium towards the complex required for catalysis[58]. For IrCp*(P1A7), pre-activation with AgOTf did indeed accelerate the reaction (1.4x faster), even in the presence of exogenous chloride (1.1x faster) in the harmaline assay, suggesting that initial chloride abstraction is the key energetic barrier for this catalyst. On the contrary, in the same assay, IrCp*(P4A11) did not benefit from pre-activation in both water (no change) and PBS (1.4x slower), suggesting that a different part of the mechanistic cycle is the turnover-limiting step. We envisage that these catalysts could be applied to molecular uncaging in bacteria or even in mammalian cells[43,56,59], but further mechanistic analysis and assay development would be required, which is beyond the scope of this work.

Overall, we have demonstrated that these metal complex libraries can be rapidly assessed for photophysical and catalytic properties using 96 well plate-based assays under ambient conditions. We envisage that any property-screening assay that is amenable to a well plate format could be adapted to test these and other combinatorial metal complex libraries to identify ideal compounds. We are currently exploring further optimization of metal scaffolds in this area.

## Biological testing of synthesized libraries

With the main focus of our group being the exploration of transition metal complexes as antimicrobial agents, we advanced all the prepared libraries to an initial biological assessment, using a "Direct-to-Biology" approach. In a primary screening, all crude reaction mixtures were evaluated against two bacterial strains, one Gram-positive (Staphylococcus aureus) and one Gram-negative (Escherichia coli). As hit-rates against Gram-negative strains are generally much lower[60], we decided to evaluate compounds against E. coli at 50 μM and against S. aureus at 12.5 μM and 50 μM to be able to better differentiate the level of activity. Metal-based compounds have a reputation of being generally toxic, although recent reports have questioned this assumption, at least in the realm of in vitro assays[11]. We therefore tested compounds against human embryonic kidney cells (HEK293T) at 50 μM. Amongst the 192 Tz-4-P and Tz-1-MP ligands, only one reduced the average viability of HEK293T cells below 50% (M12Y3, 45 ± 7% cell viability), showing that the ligands on their own are generally non-toxic (Fig. 7A, B). None of the ligands showed any significant activity against E. coli, and only four showed growth inhibition against S. aureus: M7Y1 (50 μM), M16Y2 (12.5 μM), M16Y3 (12.5 μM), and M23Y3 (50 μM). As M16 contains the broad-spectrum antibiotic ampicillin and the reactions to form those compounds had relatively lower conversion, the observation of some antibacterial activity is unsurprising. We conducted dose-dependent testing of these crudes against S. aureus to determine minimum inhibitory concentrations (MIC) of the putative ligands and found them to be 50 μM for M7Y1 and M23Y3, while lower MICs of 12.5 μM and 3.13 μM were observed for the ampicillin containing triazole compounds. Preliminary testing on the metal complex precursors

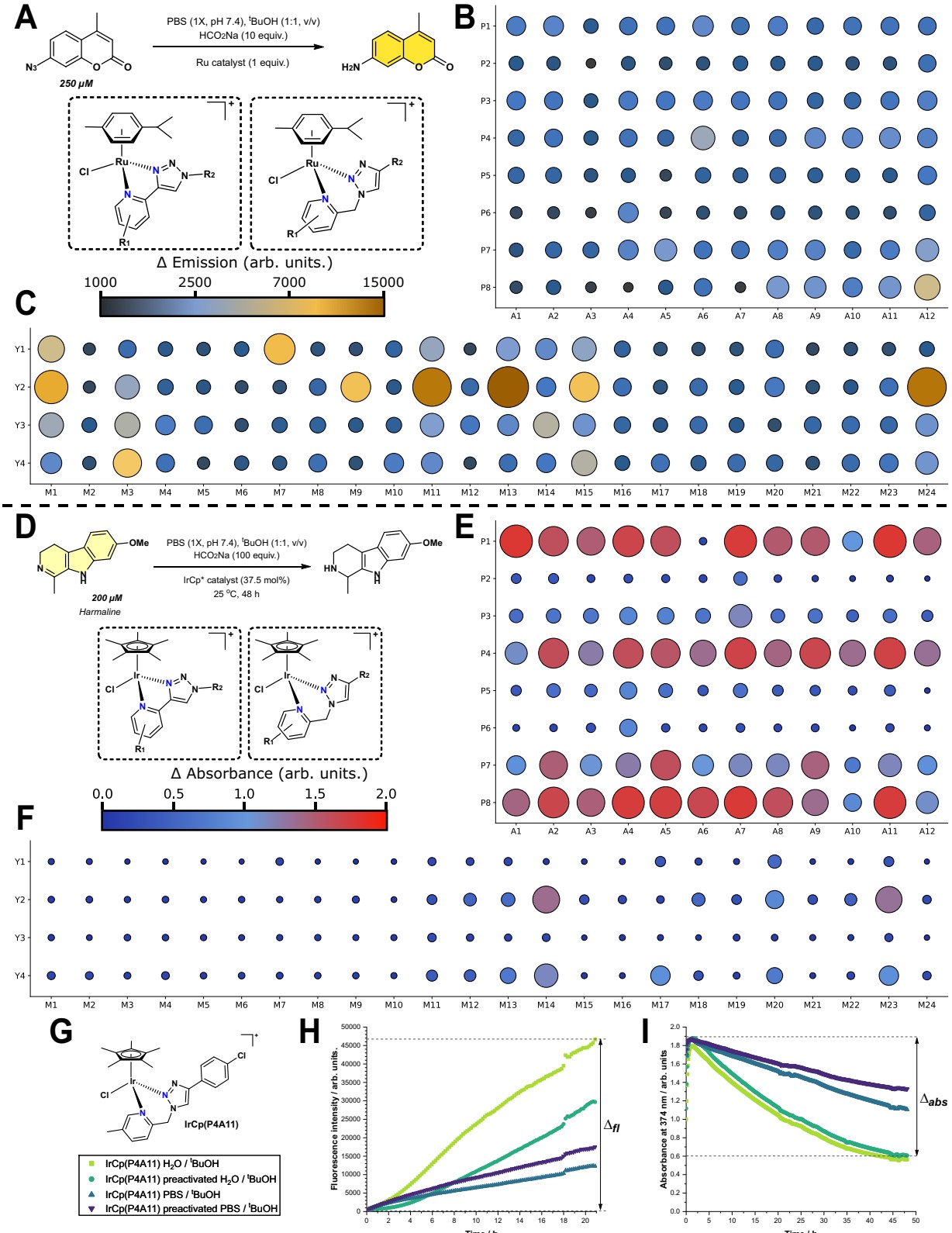

([Ir(ppy)$_2$Cl]$_2$, [Ru($p$-cymene)Cl$_2$]$_2$, Re(CO)$_5$Cl, Mn(CO)$_5$Br) showed that only [Ir(ppy)$_2$Cl]$_2$ had an MIC of 50 μM against *S. aureus.*, with the rest having a values > 50 μM. Overall, almost all the ligands showed no significant antibacterial or cytotoxic properties, and alongside the lack of activity of the metal scaffolds, this allows us to assign any biological effects observed on the metal complex crudes to the putative metal complex with high confidence.

All but one of the IrCN complexes showed antibacterial activity against *S. aureus* (Fig. 7C). However, none showed any appreciable effect on the growth of *E. coli*. This finding aligns with the general difficulty across the field of (metallo)antibiotic drug discovery to find compounds able to effectively penetrate the double-membrane and evade the efflux pumps of these pathogens[61,62]. Dose-dependent screening of the IrCN libraries against *S. aureus* revealed high

**Fig. 6 | Summary of transfer hydrogenation kinetic results for selected metal libraries. A** scheme for Ru-catalyzed transfer hydrogenation of Coumarin-$N_3$ using the RuCy libraries. Heatmaps show the change in fluorescence over 24 h for the reactions (product formation, $\lambda_{ex} = 350$ nm, $\lambda_{em} = 455$ nm), with large dark brown circles showing intensity change >15000 A.U.; **B** RuCy(Tz-1-MP) library; **C** RuCy(Tz-4-P) library. **D** scheme for the Ir-catalyzed transfer hydrogenation of Harmaline using the IrCp* libraries. Heatmaps show the change in absorbance over 48 h for the reactions (starting material consumption, $\lambda = 384$ nm), with large red circles showing absorbance change of 2.0 A.U., corresponding to total conversion.; **E** IrCp*(Tz-1-MP) library; **F** IrCp* (Tz-4-P) library. **G** Kinetic traces of transfer

hydrogenations catalyzed by IrCp*(P4A11) for (**H**) coumarin-$N_3$ assay; $\Delta_{fl}$ = the change in fluorescence between the initial measurement and the final measurement ($\lambda_{ex} = 350$ nm, $\lambda_{em} = 455$ nm); Conditions: coumarin-$N_3$ (250 μM), HCO$_2$Na (100 equiv.), catalyst loading (1 equiv.), solvent:$^t$BuOH (1:1 v/v), 21 h, 25 °C (**I**) harmaline assay; $\Delta_{abs}$ = the change in absorbance between the maximum measurement and the final measurement ($\lambda = 384$ nm); Conditions: harmaline (200 μM), HCO$_2$Na (125 equiv.), catalyst loading (37.5 mol%.), solvent:$^t$BuOH (1:1 v/v), 48 h, 25 °C. Traces and values are an average of 3 replicate reactions for each set of conditions. PBS = phosphate buffered saline, Cp* = pentamethylcyclopentadienyl.

### Table 1 | Results for catalytic testing of pure compounds

| | Conditions (co-solvent, pre-activation) | Coumarin assay ($\Delta_{fl}$) | Harmaline assay ($\Delta_{abs}$) | Harmaline assay (LCMS conversion%) |
|---|---|---|---|---|
| IrCp*(P4A11) | H$_2$O | 46,056 | 1.241 | 100 |
| | H$_2$O, pre-activation | 29,217 | 1.279 | 100 |
| | PBS 1X | 11,773 | 0.760 | 45.6 |
| | PBS 1X, pre-activation | 16,988 | 0.538 | 34.4 |
| | *Crude (PBS 1X)* | n.m. | 1.68 | n.m. |
| IrCp*(P1A7) | H$_2$O | 31,295 | 0.179 | n.m. |
| | H$_2$O, pre-activation | 38,581 | 0.246 | n.m. |
| | PBS 1X | 4280 | 0.098 | n.m. |
| | PBS 1X, pre-activation | 5433 | 0.110 | n.m. |
| | *Crude (PBS 1X)* | n.m. | 1.74 | n.m. |

All values are given as the average of three replicas. LC-MS conversions are given as peak area% at 254 nm. Pre-activation was achieved by heating stock solutions of catalyst with AgOTf (2 equiv.) in DMSO:H$_2$O for 1 h at 60 °C.

*n.m.* not measured, $\Delta_{fl}$ change in fluorescence (arbitrary units), $\Delta_{abs}$ change in absorbance (arbitrary units).

antibacterial activity. The lowest MIC measured was 0.39 μM for four compounds which is comparable to the standard-of-care antibiotic vancomycin (MIC = 0.42 μM), which was used as a control. Overall, 59 out of 96 compounds displayed an MIC < 5 μM.

A crucial parameter of any antibiotic is its selectivity for bacterial cells over healthy human cells, often denoted by the Therapeutic Index (TI). Unfortunately, most of the IrCN also significantly reduced human cell viability at 50 μM. However, given the very high level of antibacterial activity and the observation that the inhibition of a significant portion of the compounds was in the 25–50% range, we identified several compounds with a potentially viable therapeutic index of >10.

The Re(CO)$_3$ compounds similarly showed high levels of antibacterial activity in *S. aureus*, with 61 compounds completely inhibiting bacterial growth at 50 μM or lower (Fig. 7D). The MICs were generally higher than for the IrCN library, but the compounds were consistently less toxic. Most caused less than 50% growth inhibition in the HEK293T cells, while still having MICs as low as 0.78 μM.

Comparing the Re(CO)$_3$ and IrCN libraries, there is a clear mirroring of activity for individual ligands. Generally, if a Re(CO)$_3$ complex with a given ligand is active, then the corresponding IrCN will be equally or more active. This is noteworthy as the ligands themselves are demonstrably not active and the two metal scaffolds are clearly distinct. We speculate that the arrangement of the ligand in an octahedral geometry and the occupation of the coordinating nitrogen atoms with the metal scaffold may be the cause for this.

For the Re(CO)$_3$ library, there is a correlation between its activity and conversion efficiency. The same was not observed for the IrCN library. In terms of specific trends, Y2 and Y4-derived ligands are much less active for Re(CO)$_3$ than IrCN, which can be ascribed to lower conversion efficiency in the former. Y1 and Y3-derived ligands are the most active for Re(CO)$_3$, with active compounds having similar activity in both metal libraries. Indeed, M17–22 for Y3 shows almost identical activity for both Re(CO)$_3$ and IrCN, which suggests that the identity of the metal scaffold is not necessarily important for activity. M5 (enantiopure *(S)*-1-phenyl-ethylamine)-derived ligands are the overall most active for Re(CO)$_3$, and are similarly highly active for IrCN, while there

was no or very low activity for: M2 (pendant amide, despite high conversions), M11 (phenyl-ester, which showed good activity for IrCN despite low conversions), M15 (β-lactam, poor conversions), and M23 (biaryl, also poor conversions).

By contrast, the Mn(CO)$_3$ compounds generally showed lower to no significant antibacterial properties than its congener library (Fig. 7E). Only 21 compounds showed any antibacterial activity against *S. aureus* and only 4 had an MIC < 10 μM. Nevertheless, the overall hit-rate of 21% is comparable to the hit-rate we previously observed in Mn(CO)$_3$ Schiff-base libraries of 15.3%[31]. The hit-rate is also still much higher compared to what is generally found in purely organic molecule screening campaigns (~1%)[62]. At the same time, this compound class showed similar rates of cytotoxicity to rhenium, with half of all compounds causing cellular growth inhibition of less than 50% at 50 μM, and a handful causing greater than 75% growth inhibition.

Lastly, the RuCy libraries with both Tz-4-P and Tz-1-MP ligands (Fig. 7F, G) showed the lowest level of antibacterial properties. Only a few of the RuTz-4-P compounds inhibited growth against *S. aureus* at 12.5 μM. This is in slight contrast to previous work on ruthenium *p*-cymene Schiff-base compounds, which have shown high hit-rates and significant antibacterial activity[30,38,43,63].

Structure-Activity Relationship (SAR, see Supporting Information Section 4) analysis was performed on the data gathered to gain insights into what features contribute most to activity and toxicity. A strong correlation between the retention time (RT) of the Tz-4-P library by LC-MS and calculated cLogP values for the ligands indicates that RT is a good descriptor for lipophilicity in these libraries. For the Re(CO)3 library, the active compounds tended to have a tighter range of RTs than non-active compounds, and the molecular weight (MW) was also important (low MW compounds tending to be more active). This is in line with guidance for organic antibiotics, which indicates that lighter and more polar molecules have better uptake in bacteria[64]. Conversely, a weak trend between MW and toxicity suggests that heavier compounds are less toxic. An analysis of the fraction of carbon sp$^3$ character suggested that there is an optimum window that imparts high activity, which can be rationalized by the ligand structure affecting

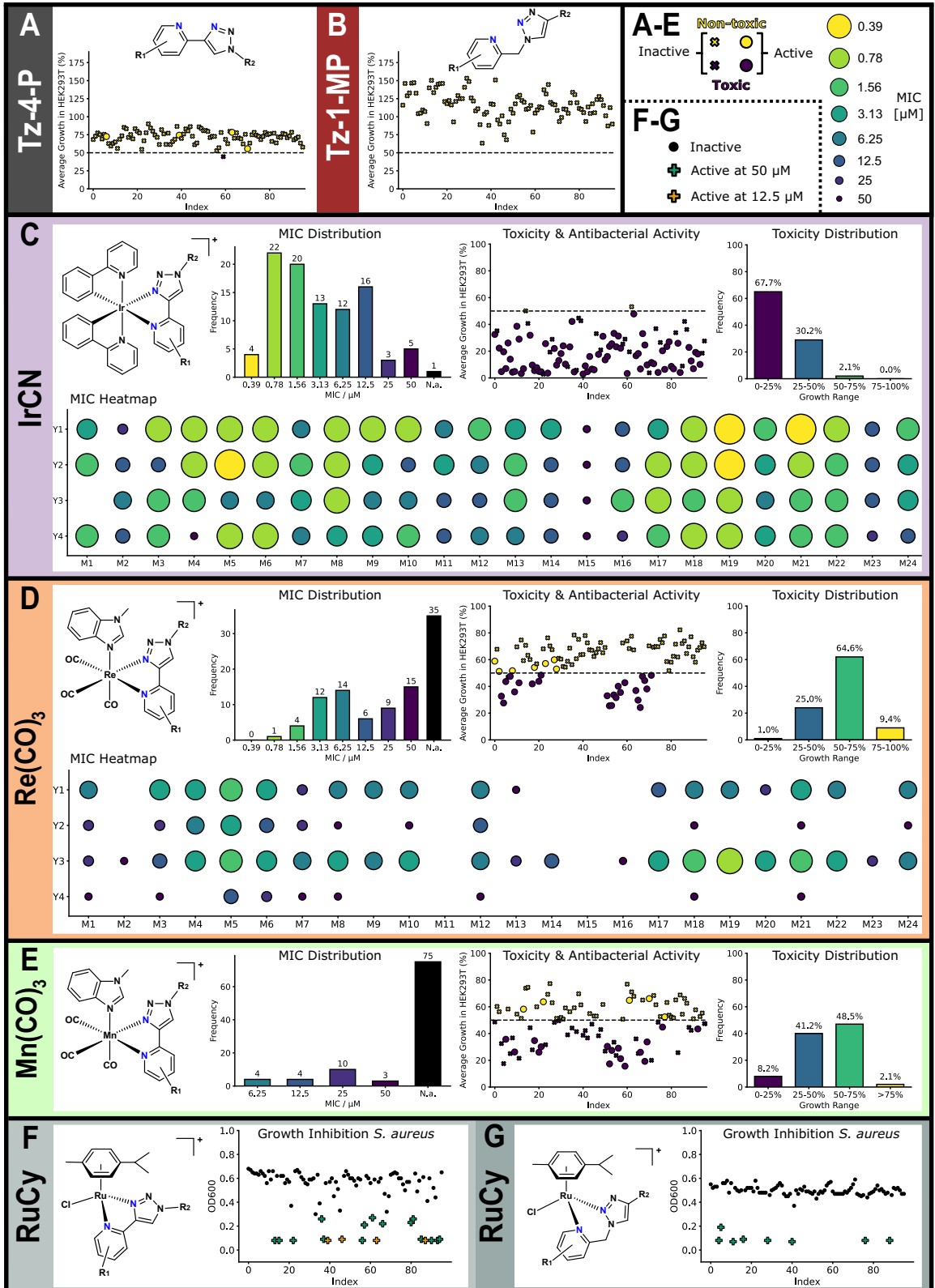

**Fig. 7 | Summary of the biological testing data on the crude ligands and metal complexes.** For **A**–**E**, average % growth of HEK293T cells refers to the toxicity of the compounds at 50 μM (single dose, 2 biological and 2 statistical repeats). Any compounds under 50% (dashed line) are labelled as "toxic". The circles on these graphs refer to whether the compounds had an MIC of ≤ 6.25 μM against S. aureus.

The MIC heatmaps show at what concentration the compounds fully inhibit growth of S. aureus (2 biological and 2 statistical repeats). For **F**, **G**, only single dose response bacterial data were collected due to low activity, with OD600 (optical density at 600 nm) referring to the concentration of bacteria present after incubation (a low value is classed as "active").

uptake into bacteria. There was also a clear correlation between the antimicrobial activity and toxicity of this library, with more active compounds being more toxic. However, at MIC 6.25 μM, most compounds are in the 50% viability range for HEK293T cells at 50 μM, so there is a good potential therapeutic window.

For the IrCN library, the trends were not as clear, but there was a weak correlation between MW and activity, with more active compounds being lighter, as well as between MW and toxicity. There is also a weak positive correlation between activity and toxicity. For a given ligand, the IrCN complex tended to be significantly more active than the same $Re(CO)_3$ compound, which was in turn more active than the $Mn(CO)_3$ complex. This suggests that for a given ligand, the identity of the metal scaffold has a larger effect on the biological activity of the resultant compound than the identity of the ligand.

Some further insights into more concrete structural contributors to antimicrobial activity in metal complexes were obtained by training a support vector machine machine learning (SVM-ML) model on the antibacterial activity data. The structures of rhenium and iridium metal complexes (which had the most interesting levels of antibacterial activity) were converted to a 598-bit ELECTRUM fingerprint based on our recent work[65]. Due to the limited dataset size of 192 compounds, we converted the antibacterial activity data into binary classification data (a compound was labelled active ("1"), if MIC value was ≤ 6.25 μM or inactive ("0") otherwise), which is commonly done even for larger datasets of antimicrobial compounds[66]. The SVM-ML model was trained with 5-fold cross validation, and the trained model showed strong performance in predicting compound bioactivity. It achieved a mean AUC of 0.83 ± 0.05, indicating good ability to discriminate between active and inactive compounds. The mean precision was 0.80 ± 0.09, demonstrating that roughly 80% of compounds predicted to be active were truly active. Similarly, the mean recall was 0.87 ± 0.06, showing that the model successfully identified the majority of active compounds. Overall, the balance between precision and recall is reflected in a mean F1 score of 0.83 ± 0.07, suggesting reliable and consistent hit prediction across cross-validation folds. This suggests that detailed molecular structural features within the compounds correlate with their biological activity. We further analyzed the feature importance through the support vectors of the SVM. Structural features that correlated to antibacterial activity against *S. aureus* included: amide bonds, fluorine substituents and higher substitution patterns around the metal coordination site (Fig. S46). Conversely, ether groups and alkyl amines correlated with inactive compounds (Fig. S47)

In summary, we have efficiently evaluated 192 ligands and 480 metal complexes for their antibacterial and cytotoxic properties (an additional 768 IrCp compounds are currently undergoing evaluation in ongoing work). This "Direct-to-Biology" methodology allows us to rapidly narrow our focus on potentially interesting compounds with a desirable therapeutic index, and the combination of single-dose response and full MIC assays on crude reaction mixtures provides key insights into activity trends across metal libraries.

## Re-synthesis and biological testing for lead compounds

Based on the MIC and single dose response toxicity data of the crude reaction mixtures, 6 complexes were chosen for re-synthesis, full characterization and re-testing (2x $Re(CO)_3$, 2x IrCN and 2x $Mn(CO)_3$). The criteria for choosing these was based primarily on antimicrobial activity, with the most active complexes identified (no compounds with an MIC higher than 6.25 μM were considered). Of these, the least toxic compounds were selected. For the IrCN library, as almost all compounds caused less than 50% cell viability, the least toxic compounds with MIC of 1.56 μM were chosen. The $Re(CO)_3$ appeared to provide a good balance of active compounds with low apparent toxicity, so again the most active and least toxic compounds were chosen. For the $Mn(CO)_3$ library, only four compounds had an MIC of 6.25 μM, so the purest compounds by LC-MS were selected for re-synthesis.

The synthesis of all compounds was achieved by first making the ligands using the scaled-up procedure for generating ligand libraries, changing the solvent from DMSO to DMF to enable a more feasible workup and purification. The ligands were then reacted with the metal in high-yielding reactions. After purification by flash column chromatography, the pure compounds were fully characterized. Single crystals of $Mn(CO)_3(M19Y1)$, $Re(CO)_3(M1Y1)$, and $Re(CO)_3(M20Y3)$ were obtained by slow vapor diffusion ($CHCl_3/Et_2O$, DCM/$n$-hexane, and DCM/$Et_2O$ respectively), with the X-ray diffraction structures (Fig. 8) clearly showing the bidentate binding mode of the pyridine-triazole ligands.

Stability assays on the pure compounds were undertaken, with biologically relevant solvents and mixtures tested (DMSO, $H_2O$, 1X PBS and a combination of DMSO with $H_2O$ and 1X PBS). The stability was ascertained by measurement of UV absorbance (320 nm) at 37 °C over 21 h. Most compounds demonstrated good overall stability with less than 10% difference in the spectra over this period; however, the $Mn(CO)_3$ complexes exhibited significant changes in their overall UV-Vis spectra in $H_2O$ and 1X PBS over 2–8 h. This could be due to anion exchange (in the case of PBS) or water competitively binding with the complex in the axial position. However, this did not seem to be detrimental to the antibacterial activity of these compounds (*vide infra*).

Biological studies were undertaken on the purified compounds to confirm activity against bacteria (MIC assays) and ascertain toxicity ($CC_{50}$ measurements against HEK293T cells). Almost no activity was observed against *E. coli* at the measured concentrations, with the exception of $Re(CO)_3(M1Y1)$ and $Mn(CO)_3(M22Y1)$ (MIC = 100 μM, see Supporting Information Table S5). However, perturbation of the outer membrane by polymyxin B nonapeptide (PMBN) resulted in significant activity against *E. coli* for all compounds (see Supporting Information Table S6)[67]. PMBN has no inherent antibacterial activity, but damages and weakens the outer membrane of Gram-negative bacteria by interacting with the anionic lipopolysaccharide molecules. This permeabilizes the membrane to antibiotics, suggesting that uptake into Gram-negative bacteria is the issue for these compounds and they are not necessarily Gram-positive specific[68].

For the Gram-positive strains, high activity was observed in almost all cases. A comparison of activity between the crude reaction mixtures and re-synthesized compounds against *S. aureus* gives an increase in activity in all cases, with very low MICs (5/6 compounds active in the nanomolar range, similar to the vancomycin control MIC = 0.42 μM, Table 2). This confirms that in all cases it is the putative metal complex in the crudes that is responsible for the observed activity, validating our methodology of pre-screening large crude libraries (Table 2). It should be noted that in several cases ($Re(CO)_3(M1Y1)$, IrCN(M12Y1), $Mn(CO)_3(M19Y1)$) there were larger increases ( > 16x, >4x, and >8x more active respectively) in antibacterial activity than expected from the increase in purity due to re-synthesis (crude purity by LC-MS of 43%, 72%, 41% respectively). This suggests that our methodology in fact underestimates the activity of the complexes in the synthesized libraries.

The other tested Gram-positive strains showed generally good activity, with slightly higher MICs against *Enterococcus faecalis* in most cases. The exception was the manganese complexes, which were significantly less active. *Enterococcus faecium* showed very low growth in the Mueller-Hinton (MH) growth medium that was typically used, so Tryptic Soy Broth (TSB) growth medium was used instead. This gave measurable MICs, but a comparison with *E. faecalis* (which was tested in both MH and TSB) indicates that TSB results in a significant rise in MIC values. As such, although the MICs for *E. faecium* are higher than *S. aureus*, there is still good activity in most cases (Table 2). Generally, $Re(CO)_3(M20Y3)$ and IrCN(M12Y1) show exceptional activity across the tested Gram-positive (and sensitized Gram-negative) bacterial strains, with often nanomolar MICs. The fact that the compounds showed very low MICs against *E. coli* upon permeabilization suggests that further structural optimization targeted at improving Gram-negative bacterial

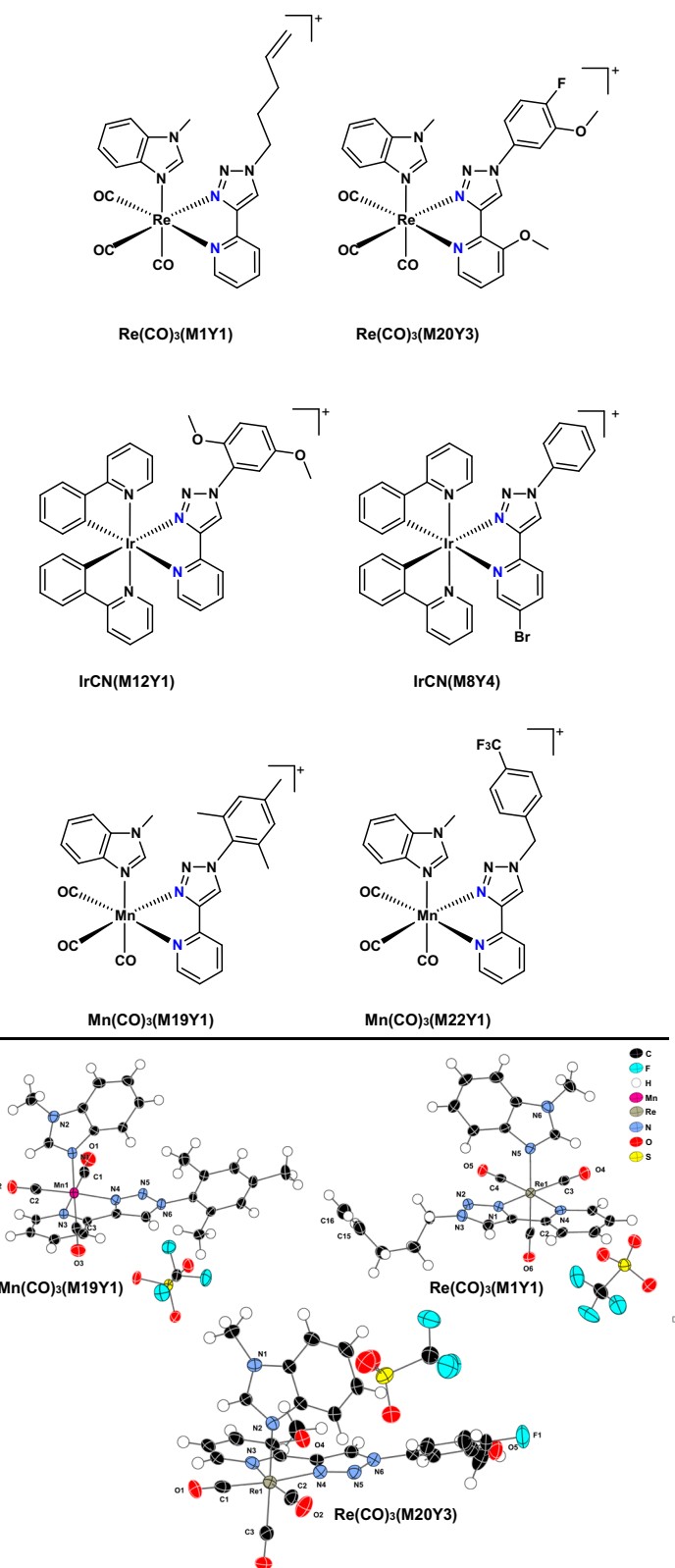

**Fig. 8 | Structures of selected lead complexes.** (Top) Chemical structures of lead complexes for re-synthesis and biological testing (MIC and toxicity); (bottom) single crystal X-ray diffraction structures of Mn(CO)₃(M19Y1), Re(CO)₃(M1Y1), and Re(CO)₃(M20Y3) (thermal ellipsoids set at 50% probability and hydrogens represented as spheroids). CDCC Deposition numbers 2504789, 2455425, 2504788.

uptake could result in pan-active metalloantibiotics. As the antibacterial activity of the pure compounds increased compared to the crude, the same trend was also observed for their cytotoxicity. Indeed, most of the compounds displayed CC₅₀ in the range of 10 μM.

However, due to their high level of antibacterial activity, all compounds except Mn(CO)₃(M22Y1) still displayed a favorable therapeutic index >10. The best ratio of antibacterial activity and toxicity was determined for IrCN(M8Y4) showcasing a therapeutic index between

**Table 2 | Biological data of lead compounds**

| | MIC [µM] vs. bacteria | | | | Toxicity [µM] | | | |
|---|---|---|---|---|---|---|---|---|
| | S. aureus | | E. faecium[a)] | E. faecalis | HC$_{10}$ | Crude toxicity (%viability, 50 µM) | CC$_{50}$ | TI |
| | Crude | Pure | | | | | | |
| Re(CO)$_3$(M1Y1) | 6.25–12.5 | 0.39 | 12.5 | 1.56–3.13 | 73 ± 29 | 59 ± 8 | 12.4 ± 1.2 | 32 |
| Re(CO)$_3$(M20Y3) | 3.13–6.25 | 0.39–0.78 | 3.13 | 0.78–1.56 | 36 ± 6 | 47 ± 8 | 8.2 ± 0.7 | 11–21 |
| IrCN(M12Y1) | 1.56–3.13 | 0.20–0.39 | 3.13 | 1.56 | 81 ± 4 | 29 ± 3 | 7.3 ± 1.2 | 19–37 |
| IrCN(M8Y4) | 1.56–3.13 | 0.39–0.78 | 6.25 | 1.56–3.13 | >200 | 27 ± 6 | 38.5 ± 5.5 | 49–99 |
| Mn(CO)$_3$(M19Y1) | 6.25 | 0.39–0.78 | 12.5 | 3.13–6.25 | >200 | 35 ± 12 | 11.9 ± 1.1 | 15–31 |
| Mn(CO)$_3$(M22Y1) | 6.25 | 3.13–6.25 | 25 | 12.5–25 | 87 ± 7 | 30 ± 18 | 12.9 ± 1.2 | 2–4 |
| Vancomycin [µg / mL] | 0.625 | 0.625 | 0.625 | 0.313 | | | | |

Minimum Inhibitory Concentration (MIC) and toxicity data for the re-synthesized compounds against a selection of Gram-positive strains. Antibacterial activity is displayed as MIC [µM]. A "-" indicates no activity up to 100 µM; HC$_{10}$–human red blood cells, the concentration at which 10% of cells have lysed; CC$_{50}$–HEK293T cells the concentration at which 50% of the cells are not viable, the error recorded is the standard deviation; TI–therapeutic index, determined by dividing the lowest value between CC$_{50}$ and with the lowest MIC value for each compound against S. aureus; MIC determined with n = 3 across two biological replicates.
[a]TSB growth medium used.

49–99. Overall, these values are comparable and in several cases superior to that of other recently reported metal-based antibiotics[16,23,31].

Additional toxicity testing in the form of haemolysis was undertaken. For human red blood cells, the concentration at which 10% of them have lysed (HC$_{10}$) was low to moderate for the two Re complexes (HC$_{10}$ = 36 ± 6 µM and 73 ± 29 µM, respectively), mirroring their toxicity against mammalian cells. This was also true for IrCN(M12Y1) and Mn(CO)$_3$(M22Y1), which also caused haemolysis, albeit at higher concentrations (HC$_{10}$ = 81 ± 4 µM and 87 ± 7 µM respectively). Interestingly, both IrCN(M8Y4) and Mn(CO)$_3$(M19Y1) showed no haemolysis at any of the tested concentrations (HC10 > 200 µM). For IrCN(M8Y4), this is especially encouraging, as it is highly active against Gram-positive strains, has the highest CC$_{50}$ tested and is not haemolytic. For all compounds, the HC$_{10}$ values are significantly higher than the MIC concentrations against Gram-positive strains, so there is still a potential therapeutic window available.

From a methodology perspective, the single dose toxicity assay was used as a guide to select the most active and least toxic compounds for re-synthesis. It is clear from the CC$_{50}$ data that although the single dose assay mirrors the true toxicity, it can underestimate the cytotoxicity of the pure compounds. For the IrCN compounds, it was expected that they would have significant toxicity based on recent work, and the single dose assay did indeed indicate that[22]. More surprising is the Re(CO)$_3$ data, which was significantly more toxic than expected from the preliminary assay. As such, the high throughput assay does successfully indicate toxicity, but more stringent conditions on selecting active and non-toxic compounds will be used in future work, such as increasing the toxicity threshold above 50% cell viability. It should be noted that cell-based assays can only provide so much information and hit compounds will have to undergo more stringent testing before a conclusion can be made about their safety.

In summary, we have presented a combinatorial methodology to generate large libraries of pyridyl-triazole bidentate ligands formed by CuAAc Click reactions. These libraries have been demonstrated to coordinate efficiently to a number of metal scaffolds, generating 672 well-characterized complexes under mild conditions. We have showcased some applications of our metal libraries for catalytic and photophysical applications, including synthesizing and testing two iridium complexes in transfer hydrogenation assays. We have successfully screened all compounds for antimicrobial activity and toxicity in a high-throughput "Direct-to-Biology" approach, and, using the available data, have re-synthesized several highly active metalloantibiotics, with MICs as low as 200 nM against S. aureus. We have also demonstrated that the single-dose antibacterial and toxicity assays are a reliable indicator of the properties of the pure compounds from crude reaction mixtures, even

for low-yielding reactions. However, care must be taken when it comes to single-dose toxicity assays as they tended to indicate that compounds are less toxic than they turned out to be upon re-synthesis.

We have identified one complex by our Direct to Biology approach, IrCN(M8Y4), which has good activity against Gram-positive strains and relatively low toxicity against HEK293T cells, giving it a favorable therapeutic index ( > 49 against S. aureus). Additionally, it has been shown to be non-haemolytic and demonstrates good stability in a variety of relevant solvents. These properties make it a potential candidate as an antibiotic against systemic infections. Further testing will involve advanced toxicity and stability studies, efficacy against a broader panel of Gram-positive bacteria, and potential in vivo testing.

Although promising activity against Gram-positive bacteria has been reported here for a number of different metal complexes, activity against Gram-negative strains has been lacking. Improved activity in the presence of the PMBN sensitizer suggests that the low original activity is in part due to poor uptake of the synthesized metal complexes. However, thanks to the combinatorial and versatile ligand architecture there is now scope to significantly vary the ligands to explore and include motifs to improve uptake into bacteria[69,70].

Screening crude reaction libraries for catalytic applications has also been demonstrated, with 384 complexes evaluated across two different transfer hydrogenation assays. These assays are amenable to high-throughput reaction monitoring under ambient conditions in 96-well plates and allowed us to identify and resynthesize two promising IrCp* catalysts. One of them, IrCp*(P4A11), was highly active in both assays and shows promise for further catalytic testing, including prodrug unmasking in cells. Work in our group is currently ongoing with derivatives of IrCp* and IrCN libraries to explore photophysical properties, photodynamic therapy, and catalytic applications.

The potential scope of this ligand chemistry is somewhat limited by the availability of 2-alkyne pyridines (for the Tz-4-P library). However, this limitation is more than made up for by the capacity to use practically any primary amine to generate bidentate ligands, and hence a large number of metal complexes are now obtainable. Additionally, as the ligand architecture is highly variable around a fixed core motif, this methodology lends itself well to potential machine learning applications to enable the discovery of highly optimized compounds[30]. In terms of other biological applications, assays for any disease-based screening would in principle be possible on these libraries or derivatives of them.

We believe there is much scope to expand the types of chemistry that are amenable to the combinatorial generation of ligand structures described here. Additionally, we have only scratched the surface of the periodic table in terms of explored metal scaffolds. While each metal may require slightly different conditions and not all of them will be

amenable to this type of chemistry, we believe that the field of combinatorial metal complex synthesis is only just at its beginning.

## Methods

### General equipment and software

NMR spectra were obtained using a Bruker Avance Neo 700 instrument fitted with a liquid $N_2$ cooled triple resonance cryoprobe (700 MHz [$^1$H]), Bruker AVIIIHD 600 Widebore instrument (600 MHz [$^1$H]), or JEOL ECX400 or JEOL ECS400 spectrometer (400 MHz [$^1$H]). High-resolution mass spectrometry (HRMS) ESI-MS spectra were measured using a Bruker Daltonics micrOTOF MS, Agilent series 1200LC with electrospray ionization (ESI) or on a Thermo LCQ using electrospray ionization, with <5 ppm error recorded for all HRMS samples. LCMS data was collected using a Dr. Maisch ReproSil-Pur 120 ODS-3, 2.4 μm, 50 mm×3 mm column, on a ThermoScientific Vanquish instrument, coupled to a ThermoScientific HCT ultra ETD II ion trap. The solvent system consisted of $H_2O$ / MeCN, both containing 0.1% formic acid. Characterization data and assessments of purity on the generated libraries were done using 254 nm absorbances. UV-Vis absorbance and fluorescence data were collected using a Tecan Infinite 200 Pro plate reader, using the "Greiner 96 Flat Bottom Transparent Polystyrene" option. 96-well reactions were shaken and heated using an Eppendorf ThermoMixer with "plates" attachment. Diffraction data were collected at 110 K on an Oxford Diffraction XtaLAB Synergy HyPix-Arc 100 diffractometer with Cu-K$_\alpha$ radiation (λ = 1.54184 Å). The crystals were cooled with an Oxford Instruments Cryostream 1000.

NMR data were collected using Bruker Topspin 3.5pl6 (600 MHz), TopSpin 4.1pl3 (700 MHz). LCMS data was collected using Thermo Scientific Xcalibur 4.4. High-resolution mass spectrometry data were collected using Bruker Compass HyStar 4.1. FT-IR data collected using PerkinElmer Spectrum v10.5.4. UV-Vis data, fluorescence and biological data were collected using Techan i-control 2.0. XRD data were collected using Rigaku CrysAlisPro v44.128a. Data analysis was performed for High-resolution mass spectrometry using Bruker Compass DataAnalysis. NMR data were processed using MestReNova 15.1.0-37919. XRD data were processed in Olex2-1.5-ac6-020. Biological data (CC50, Haemolysis) were analysed and processed using GraphPad Prism v8.0.2 (GraphPad Software, California, USA). Code writing was done on Visual Studio Code 1.105.1, using Python 3.11.9.

### Synthetic methods

**Preparation of azide transfer reagent.** Method adapted from Meng et al.[28] NaN$_3$ (99 mg, 1.524 mmol, 1 equiv.) was dissolved in deionized water (3.0 mL) in a 15 mL Falcon tube. To this, 3.0 mL of MTBE was added and the mixture stirred vigorously at 0 °C. Separately, 1-(fluorosulfonyl)-2,3-dimethyl-1H-imidazol-3-ium trifluoromethanesulfonate (600 mg, 1.83 mmol, 1.2 equiv.) was dissolved in 150 μL of MeCN in a Falcon tube, and was added to the reaction mixture. A further 150 μL of MeCN was used to wash the Falcon tube and ensure efficient transfer of material, with this also being added to the reaction mixture. The mixture was stirred vigorously at 0 °C with the lid loosely screwed on top for 15 min, then allowed to settle at r.t. for 30 min. The upper organic layer was decanted and diluted with 3 mL DMSO, giving a solution of FSO$_2$N$_3$ of ~173 mM.

**General procedure for synthesis of ligand libraries.** Method adapted from Meng et al.[28]. The following procedure assumes a concentration of 173 mM for FSO$_2$N$_3$ and describes the addition to 1 well of a Paradox® 96 well plate (700 μL glass vial) using an Opentrons OT2 liquid handling robot (P300 Gen2 300 μL pipette and P20 Gen2 20 μL pipette). To the reaction well, sat. KHCO$_3$ solution (11 μL, 4 equiv., nominally 3 M), followed by DMSO (65 μL), was added, then amine stock solution (100 μL, 80 mM, 1 equiv.). FSO$_2$N$_3$ stock solution was then added (48 μL, 173 mM 1.05 equiv., Note: the pipette was aspirated 3

times to equilibrate solvent vapor pressure). The reaction vials were then sealed, and shaken at 800 rpm for 18 h at r.t. Sodium ascorbate solution (80 μL, 50 mM, 0.5 equiv.) was added and the reaction shaken for a further 15 min at r.t. To each reaction well, alkyne stock solution (100 μL, 80 mM, 1 equiv.) was added, followed by (BimH)$_3$ solution in DMSO (10 μL, 20 mM, 2.5 mol%), then CuSO$_4$ solution in water (10 μL, 20 mM, 2.5 mol%). The reaction was sealed and shaken at 800 rpm for 18 h at r.t. The reaction mixture was analyzed by LCMS (5 μL in 200 μL MeCN / H$_2$O (60:40)) to confirm the presence and identity of the product ligand.

**Synthesis of IrCN(Tz-4-P) library.** IrCN dimer ([Ir(ppy)$_2$Cl]$_2$, 34.3 mg, 0.032 mmol) was dissolved in DMSO (3.2 mL, 10 mM). Ligand stock solution (Tz-4-P library, 30 μL, 20 mM) was dispensed into 700 μL glass vials, with 30 μL of IrCN dimer stock solution added to each well, followed by 60 μL of water (giving 120 μL of 5 mM complexes). The plate was sealed and heated at 60 °C (600 rpm) for 18 h.

**Synthesis of Re(CO)$_3$(Tz-4-P) library.** Re(CO)$_5$Cl (50.6 mg, 0.14 mmol) and AgOTf (36 mg, 0.14 mmol. Note: hydroscopic, light-sensitive) were placed in a 20 mL microwave vial. MeOH (7 mL) was added, and the mixture was heated in a Biotage Initiator microwave reactor at 120 °C for 30 mins. After cooling, the supernatant was used as a stock solution of [Re(CO)$_3$(MeOH)$_3$][OTf] (20 mM). Ligand stock solution (30 μL, 20 mM) and 1-methylbenzimidazole stock solution (30 μL, 20 mM, DMSO) were dispensed into 700 μL glass vials, with 30 μL of Re(CO)$_3$(MeOH)$_3$OTf stock solution added to each well. The plate was sealed and heated at 70 °C (600 rpm) for 18 h. After this, 240 μL of MeOH was added to each well, and the plate was left unsealed for one week, allowing for the MeOH to evaporate off (giving 60 μL of 10 mM complexes).

**Synthesis of Mn(CO)$_3$(Tz-4-P) library.** (Note: Mn complexes are light sensitive, so all steps were done in darkness. The pre-activation of Mn was performed under $N_2$, but other steps were done under ambient conditions). Mn(CO)$_5$Br (38.5 mg, 0.14 mmol) and AgOTf (36 mg, 0.14 mmol. Note: hydroscopic, light-sensitive) were placed in a 50 mL 2-necked round bottomed flask fitted with a condenser, and evacuated/backfilled with $N_2$ three times. Dry THF (7 mL) was added, and the mixture was refluxed under $N_2$ at 70 °C for 45 min. After cooling, the supernatant was used as a stock solution of Mn(CO)$_3$(THF)$_3$OTf (20 mM). Ligand stock solution (30 μL, 20 mM) was dispensed into 700 μL glass vials, with 30 μL of Mn(CO)$_3$(THF)$_3$OTf stock solution added to each well. After 2 h at r.t., 1-methylbenzimidazole stock solution (30 μL, 20 mM, DMSO) was added, and the plate was left shaking at r.t. (600 rpm) for 72 h, allowing for the THF to evaporate off (giving 60 μL of 10 mM complexes).

**Synthesis of IrCp*(Tz-4-P) and IrCp*(Tz-1-MP) libraries.** di-μ-chloro-bis[chloro(pentamethylcyclopentadienyl)iridium(III)] (49.4 mg, 0.062 mmol) was dissolved in DMSO (6.2 mL, 10 mM). Ligand stock solution (30 μL, 20 mM) was dispensed into a polystyrene 96 well plate, with 30 μL of IrCp* stock solution added to each well (giving 60 μL of 10 mM complexes). The plate was left stationary for 24 h at r.t.

**Synthesis of RuCy(Tz-4-P) library.** [Ru(p-cymene)Cl$_2$]$_2$ (RuCy, 21.4 mg, 0.035 mmol) was dissolved in DMSO (3.5 mL, 10 mM). Ligand stock solution (30 μL, 20 mM) was dispensed into a polystyrene 96 well plate, with 30 μL of RuCy stock solution added to each well, followed by 60 μL water (giving 120 μL of 5 mM complexes). The plate was left stationary for 18 h at r.t., before analysis by LCMS.

**Synthesis of RuCy(Tz-1-MP) library.** [Ru(p-cymene)Cl$_2$]$_2$ (RuCy, 38.3 mg, 0.0625 mmol) was dissolved in EtOH (25 mL, 2.5 mM). Ligand stock solution (60 μL, 20 mM) was dispensed into 700 μL glass vials,

with 240 μL of RuCy stock solution added to each well. The plate was sealed and shaken at r.t. (600 rpm) for 48 h, after which the EtOH was evaporated under a flow of nitrogen, and the compounds diluted with DMSO (120 μL of 10 mM compounds), before analysis by LCMS.

**General procedure for synthesis of target ligands.** $FSO_2N_3$ was synthesized as before and diluted with DMF to approx. 200 mM in 1:1 MTBE:DMF (by volume). The desired amine (0.56 mmol, 1.0 equiv.) was charged into a 50 mL polypropylene Falcon tube, followed by DMF (7 mL). Sat. $KHCO_3$ solution (0.8 mL, approx. 2.4 mmol, 4 equiv. in water) was added, followed by $FSO_2N_3$ solution (1.05 equiv.). The resulting mixture was stirred at r.t. for 18 h. Sodium ascorbate solution (5.6 mL, 50 mM, 0.5 equiv. in water) was added, and the mixture was stirred for a further 10 min. 7 mL of DMF was added, followed by alkyne (0.56 mmol, 1.0 equiv.), then $(BimH)_3$ solution (0.7 mL, 20 mM, 2.5 mol% in DMF) and $CuSO_4$ solution (0.7 mL, 20 mM, 2.5 mol% in water). The resulting mixture was stirred for a further 18 h at r.t. The crude reaction mixture was analyzed by LCMS, then 100 mL of deionized water was added. The mixture was extracted with 3:1 $CHCl_3$: isopropyl alcohol (3 × 30 mL), followed by EtOAc (2 × 30 mL). The combined organic layers were washed with 5 wt% LiCl solution (5 × 25 mL), dried ($MgSO_4$), filtered, and the solvent evaporated. The residue was redissolved in EtOAc (40 mL) and washed with more LiCl solution (3 × 20 mL). The organic layers were dried ($MgSO_4$), filtered, and the solvent evaporated to give the crude product. Purification was achieved by flash column chromatography.

**General procedure for synthesis of target Re complexes.** $Re(CO)_5Cl$ (50 mg, 0.138 mmol, 1 equiv.) and AgOTf (35.5 mg, 0.138 mmol, 1 equiv., Note: hydroscopic, light-sensitive) were charged to a 5 mL microwave vial, followed by MeOH (2 mL). The vial was sealed and heated in a microwave reactor at 120 °C for 30 min. The colorless solution was filtered through celite (with an additional 1 mL of MeOH) into a second 5 mL microwave vial containing the target ligand (0.152 mmol, 1.1 equiv.) and 1-methylbenzimidazole (BeMeIm, 20 mg, 0.152 mmol, 1.1 equiv.). The vial was sealed and heated in a microwave reactor at 120 °C for 1 h. The solvent was removed to give the crude compound as a pale yellow solid, which was purified by crystallization (approx. 2 mL DCM, layered with approx. 15 mL n-hexane, and stored at 5 °C overnight) or flash column chromatography.

**General procedure for synthesis of target Mn complexes.** All reaction vessels and containers were covered in aluminium foil and handled in the dark to prevent degradation by light. $Mn(CO)_5Br$ (91 mg, 0.33 mmol, 1 equiv.) and AgOTf (85 mg, 0.33 mmol, 1 equiv., Note: hydroscopic) were charged to a 10 mL microwave vial, which was evacuated and backfilled with $N_2$. Dry THF (4.5 mL) was added, and the vial was sealed under $N_2$ and heated at 75 °C for 45 min, after which it was allowed to cool. The pressure was released by piercing the seal with a $N_2$ balloon when cool, and 2 mL of the yellow solution (0.15 mmol Mn complex) was decanted into a second 10 mL microwave vial containing target ligand (0.155 mmol, 1.05 equiv.) and 1-methylbenzimidazole (BeMeIm, 21 mg, 0.155 mmol, 1.05 equiv.) under $N_2$. The vial was sealed and heated at 75 °C for 2 h. The solvent was removed in vacuo to give the crude compound as a yellow solid, which was purified by flash column chromatography.

**General procedure for synthesis of target IrCN complexes.** To a 100 mL round bottom flask fitted with an air condenser, IrCN dimer (40 mg, 0.037 mmol, 1 equiv.) and target ligand (0.074 mmol, 2 equiv.) were added. $CHCl_3$ (30 mL) and MeOH (10 mL) were added, and the resulting solution was stirred rapidly at 40 °C for 18 h. The solvent was removed in vacuo, and the residue was purified by flash column chromatography.

**General procedure for synthesis of target IrCp\* complexes.** To a 7 mL glass vial, di-μ-chloro-bis[chloro(pentamethylcyclopentadienyl) iridium(III)] (40 mg, 0.05 mmol, 1 equiv.) and target ligand (0.10 mmol, 2 equiv.) were added. DCM (2 mL) and MeOH (2 mL were added, and the reaction mixture stirred overnight at room temperature. The mixture was transferred to a 20 mL vial and the solvent was evaporated in vacuo. The residue was dissolved in DCM (2 mL), to which $Et_2O$ (10 mL) was slowly added to precipitate the complex. The fine precipitate was isolated on a glass sinter and washed with $Et_2O$ (3 ×20 mL). The precipitate was dissolved in DCM, collected and the solvent removed in vacuo. $Et_2O$ (2 mL) was added to the residue and dried in vacuo, yielding the target complex as a yellow solid.

**Catalytic assays**

**General procedure for Coumarin-$N_3$ catalysis assay.** Coumarin-$N_3$ (10 mg, 0.05 mmol) and sodium formate (34 mg, 0.5 mmol) were dissolved in a mixture of 1:1 tBuOH / 1X PBS solution (25 mL) to give a solution of 2 mM coumarin-$N_3$. In a polystyrene 96 well plate, 165 μL of 1:1 tBuOH /1X PBS solution was added, followed by coumarin-$N_3$ / sodium formate mix (25 μL). 10 μL of RuCy catalyst stock solution at 5 mM was added to initiate the reaction. The reaction was monitored by fluorescence (generation of Coumarin-$NH_2$) using a plate reader ($\lambda_{ex}$ = 350 nm, $\lambda_{em}$ = 445 nm) over 21 h at 25 °C, with 2 min of shaking (400 rpm) and 13 min delay between data points.

**General procedure for harmaline catalysis assay.** IrCp\* catalyst samples were diluted to a stock solution of 750 μM (50:50 DMSO/$H_2O$). Separately, stock solutions of harmaline (4.3 mg, 0.02 mmol, 2 mM) and sodium formate (136 mg, 2 mmol, 200 mM) were dissolved in mixtures of 1:1 tBuOH / 1X PBS solution (10 mL). In a polystyrene 96 well plate, 135 uL of 1:1 tBuOH /1X PBS solution was added, followed by the harmaline stock solution (20 μL, 1 equiv.), then sodium formate stock (25 μL, 125 equiv.), and finally 20 uL of IrCp\* catalyst stock solution at 750 μM was added to initiate the reaction. The reaction was monitored by UV-Vis absorbance (consumption of harmaline) using a plate reader ($\lambda$ = 374 nm) over 48 h at 25 °C, with 2 min of shaking (400 rpm) and 3 min delay between data points.

**Photophysical measurements.** 5 μL of each compound in the IrCN library (5 mM) was diluted with 200 μL DMSO. Of this, 50 μL was taken and diluted with a further 150 μL DMSO in a polystyrene 96 well plate (31.3 μM). The UV-Vis absorbance (270–600 nm, 25 flashes) and fluorescence ($\lambda_{ex}$ = 406 nm, $\lambda_{em}$ = 450–700 nm, and $\lambda_{ex}$ = 380 nm, $\lambda_{em}$ = 450–700 nm, 25 flashes) of these compounds were then measured using a plate reader.

**Biological testing.** The following bacterial strains were used for testing: *Enterococcus faecalis* CCUG 19916 T (Gram positive), *Enterococcus faecium* CUG 19434 (Gram positive), *Staphylococcus aureus* CCUG 19434 (Gram positive), *Escherichia coli* NCTC 13476 (Gram negative). Bacterial stock solutions were prepared as follows. Single bacterial colonies were grown on Tryptic Soy Broth (TSB, Gram positive) or Miller's LB Broth (LB, *E. coli*) agar. A single colony was removed and placed in the relevant growth medium (5 mL, TSB or LB) and incubated overnight at 37 °C (200 rpm). 100 μL of this overnight culture was added to 5 mL of the same growth medium, and the day culture was incubated at 37 °C (200 rpm) until the OD600 was 0.4–0.9 (~4 h). A sample of this culture was diluted with Mueller Hinton (MH) broth to OD600 = 0.022, at which point the bacterial cultures were added to plates for testing. For all testing, samples were incubated with bacteria for 16–20 h at 37 °C, after which the absorbance at 600 nm was measured using a plate reader. Antibiotic controls were Vancomycin (stock solution at 1 mg/mL in water, MIC beginning at 20 μg / mL) for Gram positive bacteria, and Polymyxin B (stock solution at 1 mg / mL in

water, MIC beginning at 20 μg / mL for *E. coli*. Unless stated otherwise, 2 biological repeats per sample were recorded, with each having 2 technical repeats. Polystyrene 96 well plates from TPP and CytoOne were used in all cases (sterile, DNA, DNase, RNase, pyrogen free). DMSO and other solvents were kept below 5% volume in the final solutions for biological testing.

**Bacterial single dose response assay.** Samples were diluted to 750 μM using DMSO/H$_2$O (to give a 50:50 stock solution). Of these stock solutions, 20 μL (for 50 μM) or 5 μL (for 12.5 μM) per sample was added to 96 well plates. MH growth media was added to each well (making up the volume to 290 μL), followed by the relevant bacteria in MH growth media (10 μL). The plates were incubated for 16–20 h at 37 °C, after which the absorbance at 600 nm was measured using a plate reader.

**Bacterial minimum inhibitory concentration assay.** Stock solutions of samples were diluted with DMSO/H$_2$O (to give a 50:50 stock solution) so that 6 uL of stock solution can be added to 300 μL of MH growth media to give an initial concentration of either 50 μM (crude compounds) or 100 μM (pure compounds). These solutions were serially diluted (150 μL into 150 μL) 7 times (crude compounds) or 11 times (pure compounds) to give solutions of 150 μL. To these, 5 μL of the relevant bacterial strain was added in MH growth media. The plates were incubated for 16–20 h at 37 °C, after which the absorbance at 600 nm was measured using a plate reader. Pure compounds were tested with 2 biological repeats per sample, with each having 3 technical repeats.For testing with Polymyxin B nonapeptide, a solution of 300 μg / mL was prepared, and 5 μL of this was added to each well immediately before the addition of *E. coli*.

**Toxicity testing**
**Cell culture.** The human embryonic kidney (HEK) 293 T cell line was cultured in Dulbecco's Modified Eagle Medium high glucose (DMEM) (Sigma Aldrich, USA) supplemented with 10% heat-inactivated foetal bovine serum (FBS), 100 μg/mL streptomycin and 100 Units/mL penicillin, and 2 mM L-glutamine. The cells were maintained in a 5% CO$_2$ environment, at 37 °C and the culture media were replaced with fresh media every 72 hours.

**Cytotoxicity studies on crude libraries.** HEK-293T cells were seeded at 10000 cells per well and incubated overnight to enable adhesion. Thereafter, the cells were treated with either the negative control (0.1%-0.28% DMSO depending on the concentration of the crude complexes) or the crude complexes at 50 μM for 24 h. To quantify cell viability, the cells were treated with the 3-(4,5-dimethylthiazol-2-yl)-2,5 diphenyltetrazolium bromide (MTT) salt (Apollo Scientific, UK) according to the procedure described by Mosmann[71]. The absorbance at 590 nm was measured using a TECAN well plate reader instrument M1000. Two biological repeats in duplicate were performed for this study, and the recorded absorbances were adjusted for the respective crudes and growth media.

**Determination of the cytotoxic concentration 50% (CC$_{50}$).** HEK-293T cells were seeded at 10000 cells per well and incubated overnight to allow adhesion. 5 mM stock solutions of pure compounds were prepared in 1:1 DMSO: PBS 1X. In a 1.5 mL 96 well dilution plate, starting concentrations of 150 μM and 200 μM of each compound were made in growth media (1 mL). 500 uL of these were serial diluted into 500 uL of media 11 times. 500 uL of media was then added to each well added to make 1 mL of each concentration (starting at 75 μM and 100 μM). 100 uL of these solutions were added into wells containing the pre-counted cells and treated for 24 h. Thereafter, the MTT assay was used to quantify viable cells and at least three biological repeats in triplicate were conducted from

which the CC$_{50}$ was determined using GraphPad Prism v8.0.2 (GraphPad Software, California, USA). The absorbance at 590 nm was measured using a TECAN well plate reader instrument M1000. 4 biological repeats in triplicate were done for this study (12 data points). For the Mn complexes, 2 biological repeats in triplicate (6 data points) were used.

**Haemolysis.** Blood samples were obtained from healthy volunteers through Hull York Medical School. The haemolysis assay was adapted from a previously reported procedure[72]. Whole blood (5 mL) was centrifuged (Thermo Scientific SL 8 R) at 4500 rpm for 1 h at 4 °C, and the plasma was discarded. The remaining human red blood cell (hRBC) pellet was washed three times with PBS 1X (pH 7.4): for each wash, PBS was added up to 15 mL, followed by centrifugation at 4500 rpm for 1 h at 4 °C. After the final wash, the cells were resuspended in PBS to a final volume of 35 mL. The stock solutions for the samples to be tested were 2.5 mM in PBS:DMSO (60:40). For the determination of HC$_{50}$ and HC$_{10}$ values, test compounds were diluted in PBS to give a concentration range from 200 μM down to 5 μM, giving a total volume of 125 μL. Dilutions were prepared in 20 μM decrements with the final points at 10 and 5 μM. Each plate included a blank medium control (PBS) and a haemolytic activity control (1% SDS in PBS). The hRBC suspension (125 μL) was incubated with the samples in PBS in a V-shaped 96-well plate for 4 h at 25 °C. After incubation, 60 μL of supernatant was carefully pipetted to a flat bottom, clear 96-wells plate. Haemolysis was measured by absorbance at 540 nm using a plate reader (TECAN M1000). The percentage of haemolysis at each concentration was determined and the HC$_{10}$ and HC$_{50}$ were calculated by inhibitor vs. normalized response fit (Prism). The minimum haemolytic concentration (MHC) was determined by visual assessment of the wells after incubation. The MHC was defined as the lowest compound concentration at which visible haemolysis (red coloration of the supernatant and/or loss of intact red pellet) could be observed compared to the PBS control. Each experiment was repeated in triplicate (3 biological and 3 statistical replicates).

**Reporting summary**
Further information on research design is available in the Nature Portfolio Reporting Summary linked to this article.

## Data availability

The authors declare that all data supporting the current findings of this study are available in the main manuscript or in the associated Supporting Information. All data are available from the corresponding author upon request. Source data are provided with this paper. This constitutes all numerical LC-MS (retention times and conversion%) and biological (Minimum inhibitory concentrations, HEK293T Growth inhibition)) data. The collated LC-MS spectra of all combinatorial libraries are supplied as Supplementary Data 1. All LC-MS, NMR, XRD and kinetic raw data can be found at https://doi.org/10.15124/728916c9-e516-4790-b75d-41cd0d91a362 and the X-ray crystallographic coordinates for the structures Re(CO)$_3$(M1Y1), Re(CO)$_3$(M20Y3) and Mn(CO)$_3$(M19Y1) reported in this study has been deposited at the Cambridge Crystallographic Data Centre (CCDC), under deposition numbers CDCC 2455425, 2504788, 2504789. These data can be obtained free of charge from The Cambridge Crystallographic Data Centre via www.ccdc.cam.ac.uk/data_request/cif. Source data are provided with this paper.

## Code availability

The Python code used in automatic LC-MS data processing is accessible at the Frei Lab GitHub repository https://github.com/TheFreiLab/FreiLab-LCMSProcessing. The code for generating the ELECTRUM fingerprints and training ML models on the Re(CO)$_3$ and IrCN data is available under https://github.com/TheFreiLab/ClickCombChem-ML.

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

## Acknowledgements

The authors wish to thank the University of York for supporting the project. Funding from the Swiss National Science Foundation in the form of an Ambizione (PZ00P2_202016, A.F.) and COST-Project Grant (IZCOZ0_220299, A.F.) is gratefully acknowledged. Thanks also to Prof. Anne-Kathrin Duhme-Klair and Niall Donaldson for their support in setting up the lab, Prof. Ronan McCarthy (Brunel) for providing the bacterial strains, CHyME at the University of York for biolab access and Dr. Lianne Willems for the HEK293T cells. Dr Ed Bergstrom and the COEMS (York) for the use of LC-MS equipment and Karl Heaton and Dr. Matthew Davy for mass spectrometry and NMR support, respectively.

## Author contributions

A.F. devised and directed the project. A.W. and C.O. performed the toxicity studies and processed the data. R.G. was the crystallographer for the Re(CO)₃M1Y1 XRD structure and assisted with the other structures. D.R.H. performed all other experimental work and processing.

## Competing interests

The authors declare no competing interests.
