## [Transparent Peer Review file · Nature Communications]

High-throughput Triazole-based Combinatorial Click Chemistry for the Synthesis and Identification of Functional Metal Complexes

Corresponding Author: Dr Angelo Frei

Version 0:

Reviewer comments:

Reviewer #1

(Remarks to the Author)

The manuscript under review aims to develop a high-throughput combinatorial click chemistry platform for the synthesis of triazole-based metal complexes, followed by their biological evaluation as antimicrobial agents. The general idea of merging modular synthetic chemistry with biological screening remains an exciting and productive avenue in drug discovery. However, the present study suffers from fundamental scientific and conceptual weaknesses that undermine its reliability, novelty, and overall significance. 1) Compound Purity and Analytical Standards: Perhaps the most critical and immediately disqualifying issue in the manuscript is the lack of purity of the compounds under investigation. The authors proceed with photophysical measurements and biological activity screening despite demonstrating that the tested compounds are of sufficient chemical purity. This is not a minor technical oversight; it goes to the heart of the study's scientific validity. As established across the scientific community and explicitly stated in the author guidelines of major publishing houses (e.g., Springer, Wiley, ACS, RSC), a minimum compound purity of 95% is required for any molecule subjected to biological assays. This requirement exists for good reason: numerous research studies have shown that even small levels of impurities can generate significant biological effects, thereby misleading the interpretation of structure–activity relationships and potentially invalidating entire studies. It is not enough to simply state that a compound was synthesized and used. Proof of purity (e.g., HPLC, elemental analysis, clean NMR spectra, and LC-MS) is absolutely essential! In this manuscript, the authors present no evidence that any of the tested compounds reach the 95% purity threshold, and in fact, the data they do present indicate that most compounds fall significantly short of this mark. As a result, the biological and photophysical data must be viewed as unreliable and cannot be used to support any meaningful scientific conclusion. Before submission to any journal, the authors must purify all compounds to the appropriate standard, provide full characterization data, and then repeat all experimental evaluations. Failure to do so will continue to render this study unsuitable for any reputable publication. 2) Synthetic Methodology and Combinatorial Strategy: The manuscript claims to offer a high-throughput, click-chemistry-based strategy for the rapid generation of a compound library. However, a closer examination of the synthetic procedures reveals a complex, multistep process that includes repeated reagent additions and sequential manipulations. This raises serious doubts about the claim that the method is combinatorial or high-throughput in practice. True combinatorial chemistry, particularly when described as “high-throughput”, implies simple, robust, one-pot procedures that minimize the number of reaction steps, maximize yield and reproducibility, and require minimal purification. The methodology described in the current manuscript does not meet these criteria. Furthermore, the click chemistry component is not exploited to its full synthetic potential. Although the CuAAC reaction is well-known for its efficiency and modularity, its use here appears limited to decorating pre-existing metal scaffolds rather than generating fundamentally new or diverse structures. This further contributes to the lack of novelty and weakens the overall impact of the synthetic approach. 3) Chemical Diversity and Novelty: A major weakness of the manuscript is the poor chemical diversity within the reported library. Nearly all of the compounds are built upon well-known and widely studied metal scaffolds, including: Rhenium tricarbonyl complexes (i.e., *Inorg. Chem.* 2019, 58, 3895–3909), Manganese tricarbonyl derivatives (i.e., *Chem. Sci.* 2024, 15, 3907–3919), Cyclometalated iridium complexes (i.e., *Angew. Chem. Int. Ed.* 2024, 63, e202401808), Ruthenium arene compounds (i.e., *J. Med. Chem.* 2014, 57, 6043–6059). Only a fraction of prior works are referenced in the current manuscript, which raises concerns about a lack of awareness of the state-of-the-art. Omitting key citations not only diminishes the scholarly quality of the manuscript, but also misleadingly implies novelty where there is little to none. Moreover, even within these established classes, the structural variations introduced are minimal. This undermines the central premise of combinatorial exploration, namely, that the library should explore broad chemical space to uncover new structure–activity relationships. Instead, the

authors present a narrow, homogeneous collection of analogs whose differences are unlikely to yield substantially different biological outcomes. Without significant scaffold innovation, ligand diversification, or new coordination geometries, the manuscript offers little that is new or original, and therefore does not meet the standards expected of high-impact publications such as Nature Communications. 4) Biological Evaluation: Limited Depth and Mechanistic Insight: While the study does report antimicrobial activity for the tested compounds, the biological data are limited to simple inhibition assays. There is no mechanistic insight, no target identification and no attempt to understand the mode of action of the compounds. This is a missed opportunity and severely limits the biological relevance and translational value of the findings. High-impact journals require more than just activity data; they demand a deeper understanding of how compounds function at the cellular and molecular level. Without such mechanistic work, even promising activity is insufficient to justify publication in a top-tier venue. Additionally, the lack of proper controls, such as comparisons with known active metal complexes or the absence of structure–activity relationship analysis, further weakens the biological conclusions. Again, the low purity of the compounds severely undermines any interpretations of the data, raising the real possibility that observed activity may be due to unknown impurities. This study raises several interesting ideas, but unfortunately fails in execution on nearly every critical metric: Lack of compound purity renders all biological and photophysical data unreliable, Synthetic methodology is not truly combinatorial or high-throughput., Minimal novelty or diversity in chemical structures with known scaffolds, and superficial biological evaluation with no mechanistic insight. In its current form, the manuscript does not meet the standards required for publication in Nature Communications or any peer-reviewed chemistry or medicinal chemistry journal. I strongly recommend that the authors revisit the core elements of their study, perform a complete overhaul of the synthesis and characterization workflow, and critically engage with the existing literature. Only then might the work have the potential to make a meaningful contribution to the field.

Reviewer #2

(Remarks to the Author)

This manuscript by Frei and coworkers provides the access to a high number of novel metal complexes, using traditional click chemistry and SuFEx chemistry. The innovation in terms of chemistry is quite limited, however, the substantial number of complexes is impressive. Although, only few selected ones have been re-synthesized.

From the point of view of this reviewer, it is a shame that more emphasis was not dedicated to catalytic properties, rather than antibacterial properties, where many challenges need to be overcome if these were to be used clinically. This includes the cost of at least some of these metals, which appear to be a limiting factor, that should be discussed. Further, a realistic discussion of the potential of such complexes as antimicrobials should be added, including stability in vivo, type of infection (e.g., systemic vs. topical), delivery (e.g., injectable, oral, creme).

Finally, both the click chemistry and the SuFEx chemistry is not discussed in sufficient detail and seminal citations are missing.

Reviewer #3

(Remarks to the Author)

The manuscript presents a combinatorial methodology for synthesizing extensive libraries of pyridyltriazole bidentate ligands via CuAAC click chemistry. These ligands were coordinated with five different metal scaffolds under mild conditions, resulting in 672 novel metal complexes. The authors have assessed the photocatalytic potential of selected libraries. Notably, the entire set of metal complexes was screened for antibacterial activity and cytotoxicity using high-throughput methods, leading to the identification of six promising compounds. These hits were resynthesized and demonstrated nanomolar-range antibacterial activity against Gram-positive bacteria. Overall, the results are compelling, and the study is well-executed and clearly presented. I support the publication of this work in Nature Communications. However, there are a few issues that need to be addressed by the authors before acceptance.

1. The manuscript employs a combinatorial approach using metals like Ir, Ru, Re, and Mn. While the rationale for most metal choices is understandable, the specific reasoning behind selecting manganese (Mn) for this study is unclear. Could the authors provide more justification for its inclusion?
2. The purity and identity of the compounds were primarily assessed by LC-MS. However, in several HPLC chromatograms, the product peak appears alongside additional peaks that may represent impurities. These impurities could potentially influence biological or catalytic activity (e.g., antibacterial effects or photocatalysis). Have the authors considered this possibility in their analysis while screening of the compounds?
3. The manuscript states that “The lowest MIC measured was 0.39 μ M for 2 compounds...” in the IrCN libraries. However, according to Figure 7C (MIC distribution), four compounds appear to exhibit this MIC value. The authors should clarify this inconsistency.
4. In the Mn(CO)₃ library, compounds M19Y1 and M22Y1 are described as the purest based on LC-MS data. Could the authors elaborate on how these were deemed the purest?
5. While cytotoxicity was assessed using HEK293T cells, no data on hemolytic activity of the hit compounds is presented. To better evaluate bacterial selectivity and potential safety, hemolytic activity should also be investigated.
6. Given the known luminescent properties of IrCN complexes, the authors could explore their application in bacterial imaging. This would help determine whether these compounds are internalized by bacterial cells.
7. The manuscript states that “Even after 18 h of irradiation, no conversion >2% was achieved” under “Catalysis.” However, Figure S4 shows a maximum conversion of only 0.54%. This statement should be corrected to reflect the actual data.
8. The catalytic study using Coumarin-N₃, while interesting, appears disconnected from the antibacterial activity. No prodrug activation has been demonstrated in bacterial cells or extracellular media. This section could be omitted or moved to the

Supporting Information to maintain focus and clarity.

9. The authors should provide possible mechanistic or structural explanations as to why M11Y2, M13Y2, and M24Y2 exhibited superior catalytic activity compared to their Ru-based analogues.

10. In many parts of the text, only the figure number is mentioned without specifying the relevant panel (e.g., A, B, or C). This is especially problematic in figures that present multiple experiments, such as Figure 6. For clarity, the authors should refer to specific panels—for example, “Figure 6(B–D)” for the transfer hydrogenation (TH) catalysis by RuCy libraries.

11. What is the rationale behind reporting absorbance values in various solvents in Table S2? While it is expected that absorbance values will vary depending on the solvent used, why are negative absorbance readings observed for M20Y3 in DMSO, M12Y1 in DMSO/H₂O (1:1) and PBS 1X, as well as for M8Y4 and M22Y1 in certain solvents? Do these negative values indicate that the complexes are unstable or degrading in these solvent systems?

Version 2:

Reviewer comments:

Reviewer #1

(Remarks to the Author)

The submitted manuscript represents a revised version of a previously evaluated study. Although the authors have provided additional details and have made certain clarifications in response to earlier reviewer comments, the major weaknesses identified in the initial submission remain unresolved. These deficiencies are not minor technical issues but rather fundamental scientific flaws that directly compromise the validity and credibility of the entire work. The most serious and immediately disqualifying issue concerns the purity of the compounds under investigation. The authors have carried out photophysical measurements and biological activity assays using samples that are demonstrably impure. In fact, based on the analytical information provided, it appears that several of the tested compounds contain impurity levels exceeding 50%. Proceeding with experimental work on such inadequately characterized materials is scientifically indefensible. Any data derived from these samples—whether photophysical, chemical, or biological—cannot be considered reliable or interpretable in a meaningful way. This issue strikes at the very core of scientific rigor and reproducibility. Across the chemical and biological sciences, it is an established and non-negotiable standard that any compound subjected to biological evaluation must possess a minimum chemical purity of 95%. This requirement is explicitly stated in the author guidelines of all major publishing houses, including Springer, Wiley, ACS, and RSC. The rationale behind this standard is clear and well-documented: even trace impurities can significantly alter observed biological effects, confounding the interpretation of structure–activity relationships and leading to erroneous conclusions. To disregard this principle is to undermine the integrity of the scientific process itself. In the present manuscript, the authors fail to provide any convincing evidence that the compounds used meet the accepted purity threshold. Essential analytical data—such as clean NMR spectra, LC-MS chromatograms, HPLC purity profiles, and elemental analysis—are either missing, incomplete, or clearly indicate that the samples are mixtures rather than pure compounds. Without proper characterization, it is impossible to assign the observed biological or photophysical properties to the intended molecules. The reported results must therefore be regarded as unreliable, non-reproducible, and scientifically meaningless. Furthermore, the authors appear to have proceeded with biological screening and photophysical measurements despite being aware of the inadequate purity of their materials. This represents a serious lapse in scientific judgment. It is the responsibility of every researcher to ensure that all data presented in a manuscript are based on rigorously characterized and validated materials. Failure to do so not only invalidates the specific findings of this work but also risks misleading future researchers and wasting scientific resources. Before this study could be reconsidered for publication, the authors would need to undertake a complete re-evaluation of their experimental approach. Specifically, they must: 1) Purify all compounds to at least 95% chemical purity, verified by appropriate analytical methods. 2) Provide full characterization data, including high-quality NMR spectra, LC-MS and HPLC purity analyses, and elemental analysis where applicable. 3) Repeat all photophysical and biological experiments using the verified pure compounds. Only after these steps are completed would the results be scientifically interpretable and potentially suitable for publication. In its current form, however, this manuscript fails to meet even the minimum standards of scientific reliability and integrity expected in the field. The core experimental data are invalidated by the use of impure materials, and the conclusions drawn from these data are therefore unsound. For these reasons, I find the manuscript wholly unsuitable for publication, particularly in a high-impact journal such as Nature Communications. In conclusion, while the authors have made certain revisions, the fundamental flaws remain unaddressed and continue to undermine the scientific foundation of the study. The lack of compound purity invalidates the reported findings and renders the entire work scientifically unreliable. I therefore strongly recommend that this manuscript be rejected.

Reviewer #2

(Remarks to the Author)

No further comments.

Reviewer #3

(Remarks to the Author)

The authors have addressed my concerns. I am satisfied with the revision work performed by the authors. The revision works have improved the quality of this manuscript. I recommend the publication as it is.

[Note from the Editor: Reviewer #3 was also asked to assess the response to Reviewer #1. In comments to the editorial office, Reviewer #3 stated they were convinced by the response and that the manuscript is now suitable for publication.]

Reviewer 1

Reviewer Comment:

1) *Compound Purity and Analytical Standards: Perhaps the most critical and immediately disqualifying issue in the manuscript is the lack of purity of the compounds under investigation. The authors proceed with photophysical measurements and biological activity screening despite demonstrating that the tested compounds are of sufficient chemical purity. This is not a minor technical oversight; it goes to the heart of the study's scientific validity. As established across the scientific community and explicitly stated in the author guidelines of major publishing houses (e.g., Springer, Wiley, ACS, RSC), a minimum compound purity of 95% is required for any molecule subjected to biological assays. This requirement exists for good reason: numerous research studies have shown that even small levels of impurities can generate significant biological effects, thereby misleading the interpretation of structure–activity relationships and potentially invalidating entire studies. It is not enough to simply state that a compound was synthesized and used. Proof of purity (e.g., HPLC, elemental analysis, clean NMR spectra, and LC-MS) is absolutely essential! In this manuscript, the authors present no evidence that any of the tested compounds reach the 95% purity threshold, and in fact, the data they do present indicate that most compounds fall significantly short of this mark. As a result, the biological and photophysical data must be viewed as unreliable and cannot be used to support any meaningful scientific conclusion. Before submission to any journal, the authors must purify all compounds to the appropriate standard, provide full characterization data, and then repeat all experimental evaluations. Failure to do so will continue to render this study unsuitable for any reputable publication.*

Response:

The reviewer states that ‘a minimum compound purity of 95% is required for any molecule subjected to biological assays’ and that ‘even small levels of impurities can generate significant biological effects’. We agree that for final compounds such a degree of purity is certainly warranted and indeed we have provided ¹H-¹³C NMR, HR-MS and LC-MS data to prove the purity for all resynthesized hit-compounds in our manuscript and SI.

However, to provide this for all compounds reported in this study is impractical and would defeat the purpose of our approach. The goal of the study was to access a large number of compounds quickly and evaluate them efficiently for a desired property. If we had set out to purify and fully characterize all compounds this would take an unfeasible amount of time. Additionally, most compounds will not have ideal properties so most of that time would be wasted.

The approach we are following is not novel and has been widely applied in both academia and industry for decades, with it recently having been referred to as “Direct-to-Biology” (D2B). Indeed, there are dozens of high-impact papers published in established journals. We, and others across both academia and industry, have shown in numerous studies that crude compounds can indeed serve as proxies to identify compounds with promising properties and that the readouts in the assays of our work for purified compounds align very well with the crude mixtures. We have collated a list of high-impact journal publications in which the same or a similar approach was taken, some of which can be found in prestigious Nature journals (a selection of these have been added to the revised manuscript):

1. Yan, K.-N. *et al.*, *Adv. Sci.*, **11**, 2400594 (2024). <https://doi.org/10.1002/advs.202400594>; High-throughput synthesis of a PROTAC library using photo-click chemistry, followed by biological testing on the crude reaction mixtures against MDA-MB-231 and MDA-MB-468 TNBC cells.
2. Leggott, A. *et al.*, *Chem. Commun.*, **56**, 8047–8050 (2020) [10.1039/D0CC02361B](https://doi.org/10.1039/D0CC02361B); Synthesis and biological testing of 220 Pd-catalysed reactions. The crude reaction mixtures were tested against S. aureus ATCC29213, and hit compounds were resynthesized.
3. Thomas, R. P. *et al.*, *Chem. Sci.*, **12**, 12098–12106 (2021) <https://doi.org/10.1039/D1SC03551G>; Synthesis and screening on crude reaction mixtures

- of reactive fragments in a direct to biology approach. Screening against human carbonic anhydrase I.
- Homer, J. A. *et al.*, *Chem. Sci.*, **15**, 3879-3892 (2024). <https://doi.org/10.1039/D3SC05729A>; Synthesis of a library of volatile organic compounds using accelerated SuFEx click chemistry. Screening of compounds for both anti-cancer and anti-bacterial efficacy, and resynthesis of target compounds.
 - Kench *et al.*, *Angew. Chem. Int. Ed.* **63**, e202401808 (2024). <https://doi.org/10.1002/anie.202401808>; Synthesis of 91 iridium complexes and testing of the crude reaction mixtures against cancer cells. Use of photodynamic therapy, and a structure-activity relationship.
 - Wilders, H. *et al.*, *Angew. Chem.*, **137**, e202418314 (2025). <https://doi.org/10.1002/ange.202418314>; Synthesis and testing of a library of cysteine targeting reactive fragments in 384 well plates – a direct to biology approach. An initial hit was further developed into other inhibitor candidates.
 - DiLuzio, S. *et al.*, *J. Am. Chem. Soc.*, **143**, 1179–1194 (2021). <https://dx.doi.org/10.1021/jacs.0c12290>; Synthesis of 1440 Ir complexes, and measurements of their photophysical properties from crude mixtures.
 - Kaguchi, R. *et al.*, *J. Am. Chem. Soc.*, **145**, 3665–3681 (2023). <https://doi.org/10.1021/jacs.2c12971>; Synthesis of libraries of peptides (>600) and screening crude reaction mixtures against an ESKAPE panel of bacteria.
 - Stevens, R. *et al.* *J. Med. Chem.*, **66**, 22, 15437–15452 (2023). <https://doi.org/10.1021/acs.jmedchem.3c01604>; Integration of direct to biology approaches by GSK in synthesising and testing PROTACS. Crude reaction mixtures tested without purification.
 - Niemeier, F. *et al.*, *J. Med. Chem.*, **68**, 2, 1316–1327 (2025). <https://doi.org/10.1021/acs.jmedchem.4c01875>; Combinatorial synthesis of a library of 495 iron(III) complexes. Testing of crude reaction mixtures against human breast adenocarcinoma and noncancerous fibroblasts, with the lead compound resynthesized.
 - Nat. Commun.* **14**, 4299 (2023). <https://doi.org/10.1038/s41467-023-39949-6>; Interview piece in Nature Communications on new approaches in medicinal chemistry, including direct to biology approaches.
 - Yamamoto, K. *et al.*, *Nat. Commun.*, **15**, 5085 (2024). <https://doi.org/10.1038/s41467-024-49484-7>; Testing a library of 686 derivatives of natural products in a direct to biology screening against bacteria. Crude reaction mixtures were used without purification, and showed activity against resistant strains.
 - Wang, Z. *et al.*, *Nat. Commun.*, **14**, 8437 (2023). <https://doi.org/10.1038/s41467-023-43614-3>; Synthesis in a 384 well plate and phenotypic screening of 229 PROTAC crude compounds in a direct to biology approach. 19 compounds were resynthesized and tested to verify effects.
 - Mahjour, B. *et al.*, *Nat. Commun.*, **14**, 3924 (2023). <https://doi.org/10.1038/s41467-023-39531-0>; High throughput catalytic amide formation reactions performed in 1536 well plate. 1280 crude reaction mixtures tested for inhibition of SARS-CoV-2M virus in a direct to biology approach

In summary, this approach (Direct-to-Biology) is a valid, known and evolving methodology that is amenable to high-throughput screening. As such, the requirement by the reviewer to 'purify all compounds to the appropriate standard, provide full characterization data, and then repeat all experimental evaluations' is an impractical suggestion.

The core hypothesis of our work is that crude reaction mixtures of carefully designed assembly reactions can be used to quickly evaluate large libraries of compounds and identify promising hits quickly. There is of course a chance for false positives, but we address this by testing all building blocks which generally are not biologically active nor possess catalytic activity as well as resynthesizing, purifying, characterising and re-testing selected hit compounds.

Indeed, in some cases the combinatorial reactions are not fully clean. However, in the majority of cases, the target compound has formed with at least 50% conversion. In the biological testing there are four possible scenarios:

Target Compound and Impurities are active:

In this case the compound will be registered as active and if the activity level is high, it will be resynthesized and purified, in which case it will become clear if activity is from the putative compound. Given that most organic and inorganic compounds are **not** active against bacteria, we expect this scenario to only occur in a minority of all cases (Frei *et al.* *Chem. Sci.* **2020** DOI: <https://doi.org/10.1039/C9SC06460E>)

Target Compound is active, impurities are inactive:

Given that in most cases 50% or more of the crude mixtures is the target compound, the antibacterial assays will still pick up the activity of the target compound and any activity observed will be due to the latter. If the activity level is high, resynthesis and purification will verify this.

Target Compound is inactive, impurity is active:

This would constitute a real false positive. If the impurity has strong activity this could lead to the resynthesis of a compound that then shows no activity. This has not occurred in our work so far (Scaccaglia *et al.*, *Chem. Sci.*, 2024, **15**, 3907-3919; [d3sc05326a](https://doi.org/10.1039/D3SC05326A); Orsi *et al.*, *Angew. Chem. Int. Ed.* 2024, **63**, e202317901; <https://doi.org/10.1002/anie.202317901>) and as such we deem it unlikely. Additionally, *if* a singular impurity showed very high activity that could be an interesting compound to further investigate as well.

Target Compound and impurities are inactive:

Based on historical data (most molecules are inactive), this is the most common scenario, and in this case the compound is not of interest, so we do not miss any of these compounds.

To better clarify our approach, we have added a paragraph supporting the “*Direct-to-biology*” into the manuscript.

The comments about using crudes for photophysical effects can be addressed similarly. We tested the two resynthesized (and pure) IrCN complexes for fluorescence and absorbance, and the emission spectra match the crude samples. This allows us to assume with confidence that the fluorescent species are indeed the target complexes.

We do acknowledge some of the issues with the approach in the revised manuscript but carefully explain the mitigations that can and were undertaken in our study.

Correction: To contextualize the approach that we have taken, we have included a paragraph in the manuscript introduction that explains “Direct-to-Biology”, using libraries of not purified compounds as a screening technique. This should leave no doubt as to the validity of our study and makes clear the approach we have taken:

To rapidly cover and focus in on the vast chemical space represented by this chemistry, we have turned to a “Direct-to-Biology” screening approach, which has been successfully used across academia and industry.^{22,32–41} This approach involves the high-throughput synthesis and characterization of compound libraries, followed by biological testing of reaction crudes without purification. Concurrently, the reaction components (in our case metal scaffolds and ligands) are tested to support that any observed biological activity is likely from the putative metal complex. There are potential issues with impurities having an outsized effect in these assays, so high yielding reactions, such as those typically used in combinatorial chemistry are favored.^{29–31} As a result, while it is possible to get false positives with a Direct-to-Biology approach, false negatives are rare. The final step is hit validation, where target compounds which showed promising properties in the crude-screening are resynthesized and purified, and finally re-tested to confirm their biological activity. Overall, Direct-to-Biology represents a powerful screening technique that allows for coverage of a large amount of chemical space

quickly, reduces time wasted in making inactive compounds, and funnels down to a small number of compounds with desirable properties. A similar approach can also be taken with photophysical and catalytic testing.^{42,43}

Reviewer Comment:

2) Synthetic Methodology and Combinatorial Strategy: The manuscript claims to offer a high-throughput, click-chemistry-based strategy for the rapid generation of a compound library. However, a closer examination of the synthetic procedures reveals a complex, multistep process that includes repeated reagent additions and sequential manipulations. This raises serious doubts about the claim that the method is combinatorial or high-throughput in practice. True combinatorial chemistry, particularly when described as “high-throughput”, implies simple, robust, one-pot procedures that minimize the number of reaction steps, maximize yield and reproducibility, and require minimal purification. The methodology described in the current manuscript does not meet these criteria. Furthermore, the click chemistry component is not exploited to its full synthetic potential. Although the CuAAC reaction is well-known for its efficiency and modularity, its use here appears limited to decorating pre-existing metal scaffolds rather than generating fundamentally new or diverse structures. This further contributes to the lack of novelty and weakens the overall impact of the synthetic approach.

Response:

We disagree with the reviewer’s assessment that our approach does not represent a high-throughput combinatorial approach. Our synthetic platform, as described, could readily be expanded to prepare thousands of novel complexes with minimal effort. Instead of scale, we chose to focus on breadth and prepared novel metal complexes across two ligand and five metal scaffolds.

While our reactions have three distinct steps, these can all be done in one pot and require a minimal number of liquid transfers, which we demonstrate can be handled by an accessible automated liquid handling robot (an OT2 as used in this work can be bought for less than 20,000 USD). We are unsure what the reviewer is refereeing to in saying ‘the click chemistry component is not exploited to its full synthetic potential’. We have shown that we can prepare two distinct types of triazole-base ligands from chemically diverse starting materials (including complex organic scaffolds such as ampicillin) in a 96 well plate array with good conversions and coordinate these to five distinct metal scaffolds. By nature of design, we chose metal scaffolds where we expected the coordination chemistry to work reliably as there has not been any literature precedent for the high-throughput preparation of the reported compounds. It is of course our hope that other researchers will adopt this methodology to expand the scope of the work even further.

We would like to point out that our work goes significantly beyond what has been done before. Previous work focused on individual metals and generally either applied Schiff-base type ligands or commercially available polypyridyl scaffolds. Our work both introduces two entirely new scaffolds to the combinatorial chemistry approach and applies it to five different metal scaffolds. Again, we then evaluate a series of properties of these compounds again going well beyond what has generally been done in previous work.

Reviewer Comment

*3) Chemical Diversity and Novelty: A major weakness of the manuscript is the poor chemical diversity within the reported library. Nearly all of the compounds are built upon well-known and widely studied metal scaffolds, including: Rhenium tricarbonyl complexes (i.e., *Inorg. Chem.* 2019, 58, 3895–3909), Manganese tricarbonyl derivatives (i.e., *Chem. Sci.* 2024, 15, 3907–3919), Cyclometalated iridium complexes (i.e., *Angew. Chem. Int. Ed.* 2024, 63, e202401808), Ruthenium arene compounds (i.e., *J. Med. Chem.* 2014, 57, 6043–6059). Only a fraction of prior works are referenced in the current manuscript, which raises concerns about a lack of awareness of the state-of-the-art. Omitting key citations not only diminishes the scholarly quality of the manuscript, but also misleadingly implies novelty where there is little to none. Moreover, even within these established classes, the structural variations introduced are minimal. This undermines the central premise of combinatorial exploration, namely, that the library should explore broad chemical space to uncover new structure–activity*

relationships. Instead, the authors present a narrow, homogeneous collection of analogs whose differences are unlikely to yield substantially different biological outcomes. Without significant scaffold innovation, ligand diversification, or new coordination geometries, the manuscript offers little that is new or original, and therefore does not meet the standards expected of high-impact publications such as Nature Communications.

Response:

We have certainly noted other combinatorial metal complex approaches, indeed we have published a review article on 'High-Throughput Combinatorial Metal Complex Synthesis'. ([10.1002/anie.202420204](https://doi.org/10.1002/anie.202420204)) However, **none of these previous studies on combinatorial synthesis utilize the triazole-type of ligand reported in our work.** As mentioned before, the choice of well-behaved metal scaffolds was by design because a) we wanted to ensure that ligand coordination would work efficiently across many different ligands under 96 well plate conditions and b) there was previous data to indicate that these compound classes had promising properties as catalysts, luminophore and antimicrobial compounds.

We strongly disagree with the notion that the modifications introduced into our ligands 'are unlikely to yield substantially different biological outcomes'. Indeed, this is directly contradicted by the data reported in the manuscript and in any medicinal chemistry paper. We clearly show that small changes within the ligands can result in large differences in properties and our approach enables the preparation of larger libraries which in turn make it possible to chart these properties to structural modifications.

To highlight the insights that can be gained we have expanded the Structure-Activity Analysis (SAR) in our manuscript. This involves calculating and observing trends between observed physical and biological properties with calculated parameters such as cLogP which are commonly used in analysis of drug molecules. More detail and graphs have been added to the SI to support this. To further validate that SAR insights can be gained from this approach we have trained a Support Vector Machine ML Model on the antibacterial activity of the rhenium and iridium complexes. The model results in a remarkably good fit and we were able to extract relevant substructure features that correlate with active compounds. The extended SAR and ML discussion has been added to the manuscript and the Supporting Information:

It has been possible to perform Structure-Activity Relationship (SAR) analysis on the data gathered to gain insights into what features contribute most to activity and toxicity. A strong correlation between the retention time (RT) of the **Tz-4-P** library by LC-MS and calculated cLogP values for the ligands indicates that RT is a good descriptor for lipophilicity in these libraries. For the **Re(CO)₃** library, the active compounds tended to have a tighter range of RTs than non-active compounds, and the molecular weight (MW) was also important (low MW compounds tending to be more active). This is in line with guidance for organic antibiotics, which indicates that lighter and more polar molecules have better uptake in bacteria.⁶⁴ Conversely, a weak trend between MW and toxicity suggests that heavier compounds are less toxic. An analysis of the fraction of carbon sp³ character suggested that there is an optimum window that imparts high activity, which can be rationalized by the ligand structure affecting uptake into bacteria. There was also a clear correlation between the antimicrobial activity and toxicity of this library, with more active compounds being more toxic. However, at MIC 6.25 μM, most compounds are in the 50% viability range for HEK293T cells at 50 μM, so there is a good potential therapeutic window.

For the **IrCN** library, the trends were not as clear, but there was a weak correlation between MW and activity, with more active compounds being lighter, as well as between MW and toxicity. There is also a weak positive correlation between activity and toxicity. For a given ligand, the **IrCN** complex tended to be significantly more active than the same **Re(CO)₃** compound, which was in turn more active than the **Mn(CO)₃** complex. This suggests that the identity of the metal scaffold has a larger effect on the biological activity of the resultant compound than the identity of the ligand.

Some further insights into more concrete structural contributors to antimicrobial activity in metal complexes were obtained by training a support vector machine machine learning (SVM-ML) model on the antibacterial activity data. The structures of rhenium and iridium metal complexes (which had the most interesting levels of antibacterial activity) were converted to a 598-bit ELECTRUM fingerprint based on our recent work.⁶⁵ Due to

the limited dataset size of 192 compounds, we converted the antibacterial activity data into binary classification data (a compound was labelled active ("1"), if MIC value was $\leq 6.25 \mu\text{M}$ or inactive ("0") otherwise), which is commonly done even for larger datasets of antimicrobial compounds.⁶⁶ The SVM-ML model was trained with 5-fold cross validation, and the trained model showed strong performance in predicting compound bioactivity. It achieved a mean AUC of 0.83 ± 0.05 , indicating good ability to discriminate between active and inactive compounds. The mean precision was 0.80 ± 0.09 , demonstrating that roughly 80% of compounds predicted to be active were truly active. Similarly, the mean recall was 0.87 ± 0.06 , showing that the model successfully identified the majority of active compounds. Overall, the balance between precision and recall is reflected in a mean F1 score of 0.83 ± 0.07 , suggesting reliable and consistent hit prediction across cross-validation folds. This suggests that detailed molecular structural features within the compounds correlate with their biological activity. We further analyzed the feature importance through the support vectors of the SVM. Structural features that correlated to antibacterial activity against *S. aureus* included: amide bonds, fluorine substituents and higher substitution patterns around the metal coordination site (Figure S45). Conversely, ether groups and alkyl amines correlated with inactive compounds (Figure S46).

Reviewer Comment:

4) Biological Evaluation: Limited Depth and Mechanistic Insight: While the study does report antimicrobial activity for the tested compounds, the biological data are limited to simple inhibition assays. There is no mechanistic insight, no target identification and no attempt to understand the mode of action of the compounds. This is a missed opportunity and severely limits the biological relevance and translational value of the findings. High-impact journals require more than just activity data; they demand a deeper understanding of how compounds function at the cellular and molecular level. Without such mechanistic work, even promising activity is insufficient to justify publication in a top-tier venue. Additionally, the lack of proper controls, such as comparisons with known active metal complexes or the absence of structure–activity relationship analysis, further weakens the biological conclusions. Again, the low purity of the compounds severely undermines any interpretations of the data, raising the real possibility that observed activity may be due to unknown impurities. This study raises several interesting ideas, but unfortunately fails in execution on nearly every critical metric: Lack of compound purity renders all biological and photophysical data unreliable, Synthetic methodology is not truly combinatorial or high-throughput., Minimal novelty or diversity in chemical structures with known scaffolds, and superficial biological evaluation with no mechanistic insight. In its current form, the manuscript does not meet the standards required for publication in Nature Communications or any peer-reviewed chemistry or medicinal chemistry journal. I strongly recommend that the authors revisit the core elements of their study, perform a complete overhaul of the synthesis and characterization workflow, and critically engage with the existing literature. Only then might the work have the potential to make a meaningful contribution to the field.

Response:

While we thank the Reviewer for their comments, we strongly disagree with them. Firstly, we reiterate that the focus of this work is the novel methodology for accessing large modular libraries of transition metal complexes across five different metal scaffolds. We then show that these libraries can be evaluated for their properties in high-throughput "Direct-to-Biology" assays, enabling the rapid identification of promising compounds amongst libraries of hundreds (and possibly thousands). We then further confirm these assays by completing resynthesis, purification and full characterization of the hit compounds.

Identifying the biological mechanism of action of a single compound represents a multi-year project and successful projects on individual molecules in this area regularly result in stand-alone articles in the highest tiers of journals. Secondly, we note that understanding the mechanism of action is not an FDA requirement for drug approval. What is required is for a compound to be shown to be efficacious and safe. Indeed, our best compound has a highly promising therapeutic index and is currently undergoing further studies. While it is a core interest of our group to investigate the mode of action of novel metalloantibiotics (Özsan *et al.* *bioRxiv* **2025** DOI:[10.1101/2025.05.30.657029v1](https://doi.org/10.1101/2025.05.30.657029v1)) and we will look to develop our lead compounds further, this clearly goes beyond the scope of this work.

Reviewer 2

Reviewer Comment:

This manuscript by Frei and coworkers provides the access to a high number of novel metal complexes, using traditional click chemistry and SuFEx chemistry. The innovation in terms of chemistry is quite limited, however, the substantial number of complexes is impressive.

Response:

We thank the reviewer for the positive assessment of our work.

Reviewer Comment

Although, only few selected ones have been re-synthesized. From the point of view of this reviewer, it is a shame that more emphasis was not dedicated to catalytic properties, rather than antibacterial properties, where many challenges need to be overcome if these were to be used clinically.

Response

We have significantly expanded the catalytic evaluation of our compound libraries. We have evaluated the 192 **IrCp** and **IrCp*** libraries for their transfer hydrogenation properties utilizing a distinct assay based on harmaline. We then compared the catalytic properties of the iridium with the ruthenium compounds and selected two **IrCp*** with high activity for resynthesis, purification and full characterization. The catalytic activity of these compounds was evaluated again and compared with the crudes. Additionally, we investigated the role of the chloride ligand on the catalytic activity by systematically removing it as well as testing the reaction in the presence or absence of exogenous chloride.

We believe this additional work significantly enhances the manuscript's catalytic angle and thank the reviewer for their suggestion

Reviewer Comment:

2) This includes the cost of at least some of these metals, which appear to be a limiting factor, that should be discussed.

Response:

While the cost of transition metals is higher than the average organic molecule there are many factors to consider. Firstly, the cost for the preparation of a drug is often only a minor factor in the overall cost of treatment. Secondly, the cost of transition metals is still several orders of magnitude below the cost of monoclonal antibodies. Thirdly, these compounds can generally be prepared in fewer than five steps which is not the case for organic drugs, which often require >10 synthetic steps. This increases costs through reagents, solvents and purification steps. Metal-based drugs are also already utilized in the clinics, with e.g. platinum-based anticancer drugs being administered to 50% of chemotherapy patients (Anthony *et al. Chem. Sci.* **2020** DOI: [10.1039/D0SC04082G](https://doi.org/10.1039/D0SC04082G))

Lastly, we do not envision metalloantibiotics to be a mass-market product but rather an emergency use drug which would only be utilized in hospitals when other antibiotics have failed. Given that the pipeline for novel antibiotics is currently very dry, we believe that an effective metalloantibiotic would still be far more desirable than having no weapons against multidrug resistant infections, even if it was expensive and not produced on the hundreds of tons scale.

Reviewer Comment:

3) Further, a realistic discussion of the potential of such complexes as antimicrobials should be added, including stability in vivo, type of infection (e.g., systemic vs. topical), delivery (e.g., injectable, oral, creme).

Response:

We thank the reviewer for his comments, and we have added the following paragraph to the conclusion addressing this.

We have identified one complex by our Direct to Biology approach, **IrCN(M8Y4)**, which has good activity against Gram-positive strains and relatively low toxicity against HEK293T cells, giving it a favorable therapeutic index (>49 against *S. aureus*). Additionally, it has been shown to be non-haemolytic and demonstrates good stability in a variety of relevant solvents. These properties make it a potential candidate as an antibiotic against systemic infections. Further testing will involve advanced toxicity and stability studies, efficacy against a broader panel of Gram-positive bacteria, and potential *in vivo* testing.

Reviewer Comment:

Finally, both the click chemistry and the SuFEx chemistry is not discussed in sufficient detail and seminal citations are missing

Response:

Seminal citations for the Click chemistry have been added to the introduction.

2 H. C. Kolb, M. G. Finn and K. B. Sharpless, *Angew. Chem. Int. Ed.*, 2001, **40**, 2004–2021.

3 C. W. Tornøe, C. Christensen and M. Meldal, *J. Org. Chem.*, 2002, **67**, 3057–3064.

4 V. V. Rostovtsev, L. G. Green, V. V. Fokin and K. B. Sharpless, *Angew. Chem. Int. Ed.*, 2002, **41**, 2596–2599.

Reviewer 3

Reviewer Comment:

1. The manuscript employs a combinatorial approach using metals like Ir, Ru, Re, and Mn. While the rationale for most metal choices is understandable, the specific reasoning behind selecting manganese (Mn) for this study is unclear. Could the authors provide more justification for its inclusion?

Response

We have published work on Mn-based combinatorial chemistry previously, and it has shown to be a potent antimicrobial with low associated toxicity (Scaccaglia et al., **Chem. Sci.**, 2024,**15**, 3907-3919;d3sc05326a). An example of a Mn antimicrobial is given in Introduction Figure 1. Additionally, a sentence justifying the choice of metal scaffolds with references has been added.

These scaffolds were chosen due to their previously demonstrated biological activities and their amenability to combinatorial chemistry.^{19,23,30,31}

Reviewer Comment:

2. The purity and identity of the compounds were primarily assessed by LC-MS. However, in several HPLC chromatograms, the product peak appears alongside additional peaks that may represent impurities. These impurities could potentially influence biological or catalytic activity (e.g., antibacterial effects or photocatalysis). Have the authors considered this possibility in their analysis while screening of the compounds?

Response:

The reviewer raises a valid concern of our “Direct-to-biology” approach which has been widely applied in both academia and industry (see our response to Reviewer #1). Indeed, in some cases the combinatorial reactions are not fully clean. However, in the majority of cases, the target compound has formed with at least 50% conversion. In the biological testing there are four possible scenarios (replicated from the response to Reviewer #1):

Target Compound and Impurities are active:

In this case the compound will be registered as active and if the activity level is high, it will be resynthesized and purified, in which case it will become clear if activity is from the putative compound. Given that most organic and inorganic compounds are **not** active against bacteria, we expect this scenario to only occur in a minority of all cases (Frei *et al.* *Chem. Sci.* **2020** DOI: <https://doi.org/10.1039/C9SC06460E>)

Target Compound is active, impurities are inactive:

Given that in most cases 50% or more of the crude mixtures is the target compound, the antibacterial assays will still pick up the activity of the target compound and any activity observed will be due to the latter. If the activity level is high, resynthesis and purification will verify this.

Target Compound is inactive, impurity is active:

This would constitute a real false positive. If the impurity has strong activity this could lead to the resynthesis of a compound that then shows no activity. This has not occurred in our work so far (Scaccaglia *et al.*, *Chem. Sci.*, 2024, **15**, 3907-3919; [d3sc05326a](https://doi.org/10.1039/D3SC05326A); Orsi *et al.*, *Angew. Chem. Int. Ed.* 2024, **63**, e202317901; <https://doi.org/10.1002/anie.202317901>) and as such we deem it unlikely. Additionally, *if* a singular impurity showed very high activity that could be an interesting compound to further investigate as well.

Target Compound **and** impurities are inactive:

Based on historical data (most molecules are inactive), this is the most common scenario, and in this case the compound is not of interest, so we do not miss any of these compounds.

We have added a paragraph supporting the “*Direct-to-biology*” into the manuscript (see response to Reviewer #1). Additionally, our testing of the ligands and metal scaffolds as controls helps to reduce the likelihood of false positives.

Reviewer Comment:

3. *The manuscript states that “The lowest MIC measured was 0.39 μM for 2 compounds...” in the IrCN libraries. However, according to Figure 7C (MIC distribution), four compounds appear to exhibit this MIC value. The authors should clarify this inconsistency.*

Response:

We thank the reviewer for spotting this error, this has been corrected.

Reviewer Comment:

4. *In the Mn(CO)₃ library, compounds M19Y1 and M22Y1 are described as the purest based on LC-MS data. Could the authors elaborate on how these were deemed the purest?*

Response:

Purity was primarily assessed by determining peak area% of the target peak compared to the total peak area% of the LC-MS trace at 254 nm. In some cases, with the Mn and Re libraries, the corresponding complex without axial ligand was also observed, along with free ligand. The highest purity compounds had the highest peak area% conversion, and the minimal quantity of free ligands.

Reviewer Comment:

5. *While cytotoxicity was assessed using HEK293T cells, no data on hemolytic activity of the hit compounds is presented. To better evaluate bacterial selectivity and potential safety, hemolytic activity should also be investigated.*

Response:

We thank the reviewer for their suggestion. We have determined hemolysis for all hit compounds and determined HC₁₀ values for all of them. These values have been added to Table 2 in the manuscript. Full data and details have been added to the SI.

Reviewer Comment:

6. *Given the known luminescent properties of IrCN complexes, the authors could explore their application in bacterial imaging. This would help determine whether these compounds are internalized by bacterial cells.*

Response:

We thank the reviewer for their suggestion, and we are indeed exploring the fluorescent properties of this compound class in ongoing work, including expansion to other cyclometallated ligands to further expand the scope of IrCN compounds in particular. However, we think that this goes beyond the scope of the current work.

Reviewer Comment:

7. *The manuscript states that “Even after 18 h of irradiation, no conversion >2% was achieved” under “Catalysis.” However, Figure S4 shows a maximum conversion of only 0.54%. This statement should be corrected to reflect the actual data.*

Response

The photocatalysis section was not successful and not particularly informative, so has been removed, and catalysis is instead focused solely on transfer hydrogenation. We have included new results from IrCp* libraries that were not originally part of the paper but have since been completed. We have also expanded this section to include testing on two new pure compounds for TH catalysis. Please see the response to Reviewer 2 regarding additional catalysis work.

Reviewer Comment:

8. *The catalytic study using Coumarin-N₃, while interesting, appears disconnected from the antibacterial activity. No prodrug activation has been demonstrated in bacterial cells or extracellular media. This section could be omitted or moved to the Supporting Information to maintain focus and clarity.*

Response

We appreciate the reviewer's comment. However, in light of the requests by other reviewers we have instead opted to expand the catalytic studies. As highlighted in a previous comment:

We have evaluated the 192 IrCp and IrCp libraries for their transfer hydrogenation properties utilizing a distinct assay based on harmaline. We then compared the catalytic properties of the iridium with the ruthenium compounds and selected two IrCp* with high activity for resynthesis, purification and full characterization. The catalytic activity of these compounds was evaluated again and compared with the crudes. Additionally, we investigated the role of the chloride ligand on the catalytic activity by systematically removing it and also testing the reaction in the presence or absence of exogenous chloride.*

We believe that the reported methodology can be utilized to identify metal compounds with promising catalytic properties and hope other research groups will further push the envelope in this direction.

Reviewer Comment

9. *The authors should provide possible mechanistic or structural explanations as to why M11Y2, M13Y2, and M24Y2 exhibited superior catalytic activity compared to their Ru-based analogues.*

Response

We thank the reviewer for their comments, and we have added a new and expanded section with mechanistic insight for the iridium catalysts. This was done with additionally synthesized iridium complexes.

We additionally tested the crude **IrCp*** libraries for transfer hydrogenation in an assay adapted from Miller *et al.* for evaluating the activity of iridium-based artificial metalloenzymes using harmaline (Figure **6E-G**).⁵⁴ Iridium-based TH systems tend to be the most active compared to other transition metal catalysts,⁵⁵ and for the harmaline assay this was demonstrated to be the case. Preliminary testing with the **RuCy(Tz-1-MP)** library showed no activity, while several of the **IrCp*(Tz-4-P)** (Figure **6G**) complexes showed high conversions of the harmaline starting material to product (measured by a change in absorbance (Δ_{abs}) at 384 nm corresponding to harmaline consumption, confirmed by LC-MS, Figure S10-11). In the cases where good conversions were observed, the electron donating methyl and methoxy groups (**Y2**, **M14**, **M23**) as well as bromine groups were present (**Y4**). This is in line with previous work in the field where small changes in ligands can have dramatic effects on the activity of the catalyst.⁵⁶ This can be ascribed to a complicated interplay of steps in TH, which first require abstraction of the coordinating chloride, formation of a hydride species (favored by electron rich ligands)⁵⁷, substrate coordination (aided by labile ligands, often electron poor)⁵⁸ and concurrent transfer of hydride (enhanced by electron rich ligands). Upon testing the **IrCp*(Tz-1-MP)** library (Figure **6F**), it became apparent that these catalysts were significantly more active than their **Tz-4-P** counterparts. As 6-membered chelates, they are more labile while remaining electron donating, which could explain the higher catalytic activity.^{47,48} Additionally, the **IrCp*(Tz-1-MP)** library tended to have a greater proportion of MeCN adduct (M^{2+}) peaks in the LC-MS traces than the **IrCp*(Tz-4-P)** library, indicating that halide abstraction is easier.

Reviewer Comment

10. In many parts of the text, only the figure number is mentioned without specifying the relevant panel (e.g., A, B, or C). This is especially problematic in figures that present multiple experiments, such as Figure 6. For clarity, the authors should refer to specific panels—for example, “Figure 6(B–D)” for the transfer hydrogenation (TH) catalysis by RuCy libraries.

Response

We thank the reviewer for their comment, and have corrected this throughout the manuscript.

Reviewer Comment:

11. What is the rationale behind reporting absorbance values in various solvents in Table S2? While it is expected that absorbance values will vary depending on the solvent used, why are negative absorbance readings observed for M20Y3 in DMSO, M12Y1 in DMSO/H₂O (1:1) and PBS 1X, as well as for M8Y4 and M22Y1 in certain solvents? Do these negative values indicate that the complexes are unstable or degrading in these solvent systems?

Response

This is to demonstrate stability in the common solvent systems. The small changes (both positive and negative) demonstrate a good overall stability, and selected UV traces are included to demonstrate no change in the absorbance profile, which would be indicative of a structural change. The exception here are the two Mn complexes, which do exhibit a change in overall profile in SI Figure S19, S20. This is mentioned in the manuscript and indicates a small degree of water sensitivity.